# Multi-Modal Foundation Models Induce Interpretable Domain-Specific Molecular Graph Languages

## Abstract

Recently, domain-specific languages (DSLs) for molecular generation have shown advantages in data-efficiency and interpretability. However, constructing such a DSL traditionally requires human expertise, whereas algorithmic construction techniques have yet to demonstrate a comparable level of quality. MMFMs have also demonstrated zero-shot capabilities across vision and text domains, but they have yet to transfer these capabilities to the graph modality. We harness their capabilities for molecular DSL induction through an unconventional solution. We render the molecule as an image, prompt MMFM to describe it as text, then use prompt learning techniques to encourage the MMFM to be consistent across both modalities. We ease the MMFM's task considerably by casting the DSL construction into an equivalent problem of constructing a tree decomposition for the molecular graph. The MMFM only needs to do a series of choice selections, replacing traditional heuristics within the tree decomposition algorithm. This enables the smooth integration of its prior knowledge without overstepping the limits of the soundness of the algorithm. For each run, we collect the MMFM's reasoning for each selection into an overall story, then have agents serve as the judge for its correctness and persuasiveness. Our method, Foundation Molecular Grammar (FMG), demonstrates significant advantages in synthesizability, diversity, and data-efficiency on challenging molecule generation benchmarks. Moreover, its compelling chemical interpretability offers built-in transparency over the molecular discovery workflow, paving the way for additional oversight and feedback.

## 1 Introduction

Domain-specific languages are the foundation to design across many scientific and engineering domains. Across many applications, DSLs are meticulously crafted by human experts who have to consider a multitude of factors, from domain-specific abstractions, practical constraints, to user considerations. Being able to construct a new, high-quality DSL on-demand for specific domains like polymers or materials science, where resources are scarce, could significantly accelerate design iteration and discovery processes. The design of new functional drugs and materials is poised to have a significant impact on our future and has gained a lot of attention within the machine learning community. However, some class-specific domains have as few as 10-20 examples, and realistically it's hard to expect domain experts to collect more than a few hundred examples at a time. There has been a large number of molecular generative models proposed in recent years. While they can achieve impressive performance when given sufficient resources, the core assumption of these approaches is access to a large amount of training data needed to first reproduce the training distribution before learning to generate new ones. This assumption is not realistic for class-specific domains, and they struggle in data-efficient settings requiring domain expertise. Domain experts also have an easier time trusting models which are interpretable, and may be more inclined to experimentally validate the outputs if they can explain the generation procedure. Traditionally, DSLs check these boxes by consolidating chemical knowledge into a form which can be scrutinized and edited while also serving as a generative model. However, writing these DSLs requires a lot of time and domain expertise. As a result, they have been given up in favor of data-driven approaches with the rise of larger labeled molecular datasets. Nonetheless, the appeal of having a compact, composable

and interpretable DSL over a black-box generative model remains the same. In a surprising turn of events, modern FMs have demonstrated impressive generalist reasoning capabilities in zero-shot settings, particularly with chain-of-thought and related techniques (Brown, 2020; Wei et al., 2021; 2022; Wang et al., 2022). FMs have also been studied for their potential to assist in the traditional design workflow (Makatura et al., 2023). This paradigm shift is open-ended and seeks to exploit the inherent knowledge and common sense reasoning abilities for a variety of tasks, including translating text to design specifications, creating design variations, and searching for designs predicated on performance. However, the aforementioned applications assume access to an existing DSL, while the task of crafting a high-quality DSL is rarely explored at all. Our work serves as the missing link. We explore the potential of FMs to craft this DSL without human intervention. We believe crafting a DSL can be itself a beneficiary of the vast compilation of knowledge used to train FMs, and we integrate MMFMs as a module within a sound framework for molecule DSL induction.

## 2    RELATED WORKS

### 2.1    LEARNING MOLECULAR GRAMMARS

Since the adoption of digital representations like SMILES, a number of grammar-based generative models have been created (Dai et al., 2018; Nigam et al., 2021; Krenn et al., 2020; Kajino, 2019; Guo et al., 2022a). In all cases, the grammar is nearly always written manually or created algorithmically, without considering the chemical validity and interpretability. (Guo et al., 2022b) tries to optimize the graph DSL construction process indirectly by parameterizing the hyperedge potential function, which controls which edges are sampled for contraction, thereby indirectly affecting the construction of the DSL. At each iteration, the agent is optimized to reinforce metrics like diversity and synthesizability evaluated on a batch of generated samples. However, this approach defeats the point of DSL crafting, which should also focus on the DSL's intrinsic qualities rather than only fitting to task-specific metrics, not to mention reinforcing evaluation metrics is essentially "validating on the test set". Another concern is that the sampling agent's predictions are also not explainable, and the chemical interpretability of the method remains unclear. (Sun et al., 2024) instead prioritizes quality and interpretability by advocating to integrate expert annotations within a graph grammar learning pipeline, but its quality is contingent on experts, limiting its generalizability. Our approach, by contrast, requires no human involvement and optimizes for the intrinsic quality of the DSL as judged by non-expert LLM agents. We use an innovative technique of saving the chain-of-thought reasoning steps for creating "design narratives", which are both interpretable artifacts of the DSL induction and surrogates for the quality of the DSL.

### 2.2    LARGE LANGUAGE MODELS AND DSLs

The interplay between LLMs and DSLs is a closely related research topic. Most problems in this area assume a given DSL and aim to translate a specification (natural language, example, etc.) into a program of the DSL. (Wang et al., 2024) finds that prompting the LLM to perform chain-of-thought by generating a specialized DSL as an intermediate step is helpful for in-context learning. However, the specialized DSL is still a subset of a given DSL, and the intermediate steps within the examples are derived by first parsing example demonstrations according to the given DSL. We adopt an existing technique which observes crafting a specialized graph DSL reduces to the problem of decomposing the graph. Although our goal is to output a DSL, we don't directly decode a DSL, since the DSL of the DSL itself can be highly constrained. We bypass the issue of decoding and instead leverage the zero-shot knowledge of MMFMs to *assist* in a fundamentally sound DSL construction procedure, where the MMFM only has to select amongst a set of operations at each step.

### 2.3    FOUNDATION MODELS FOR MOLECULAR GENERATION

Foundation Models have been trained across various domains, including language, speech, and vision. Active research is exploring their potential for molecular design (Liu et al., 2023b; Guo et al., 2023; M. Bran et al., 2024). Molecules, represented as graph data, pose challenges for existing foundation models trained on text and images. To address this, significant efforts focus on converting graph data into tokens understandable by these models (Liu et al., 2023b; Guo et al., 2023; M. Bran et al., 2024), often using notations like SMILES (Weininger, 1988). However, string-based

notations like SMILES or SELFIES are mainly for representation purposes and can lead to issues in the context of generation, such as one molecule having multiple SMILES representations. This may hinder LLMs' understanding as they lack sufficient pre-training on these notations compared to SMILES, as shown in the recent study (Guo et al., 2023). Another research avenue focuses on developing domain-specific foundation models for molecular generation (Liu et al., 2023a; Su et al., 2022; Liu et al., 2023c). These models use graph neural networks (GNNs) for molecules and million-parameter language models for text, which are less powerful than LLMs. Besides, aligning these LMs and GNNs requires extensive training resources. Aware of these challenges, our work explores an alternative route, by rendering molecules as images alongside self-generated textual descriptions, implicitly aligning the two modalities at inference time. This comes at a ripe opportunity when cheminformatics APIs like RDKit are becoming prevalent enough that MMFMs are likely to have seen sufficient examples of the API during pretraining. Our Appendix case studies show MMFMs like GPT-4o can identify and reason about substructures present in rendered images of a molecule with near perfect accuracy, as judged by a real expert.

## 3 METHOD

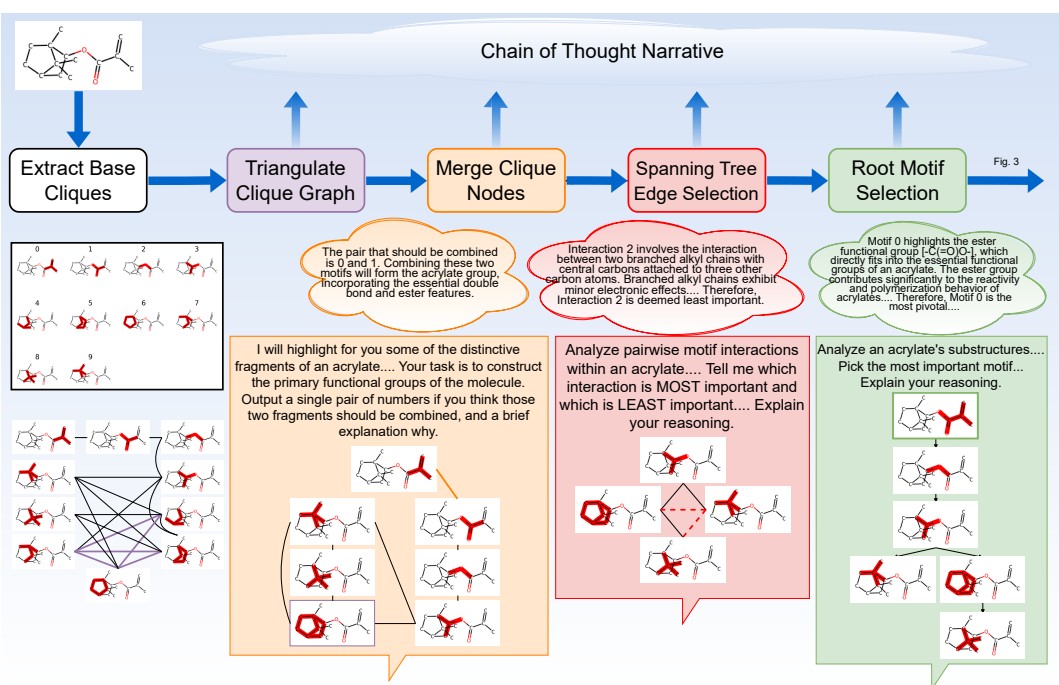

Figure 1: Main modules of FMG algorithm (left) we initialize base cliques using bonds and minimal rings, (left-middle) we triangulate the clique graph to guarantee existence of a clique tree, (middle) we prompt MMFM to meaningfully merge pairs of motifs, (middle-right) we eliminate cycles in the clique graph by prompting MMFM to identify the least important interactions, (right) we prompt MMFM to select the root motif, completing the tree.

FMG combines the sound framework of the clique tree decomposition algorithm with the adaptability of MMFM decision-making modules. FMG formulates DSL induction as constructing a clique tree, and serializes the construction into intuitive selection steps for the MMFM module to follow. In Fig. 1, we see a concrete example for an Acrylates. The algorithm first initializes most basic units – the base cliques – then proceeds to hand over control to the MMFM's selection modules. The MMFM can merge the base cliques to form chemically meaningful substructures (3.3.1 and 3.3.2), remove connections between cliques in the process of spanning tree construction (3.3.3), and finally selecting a root motif to anchor the parse tree (3.3.4).

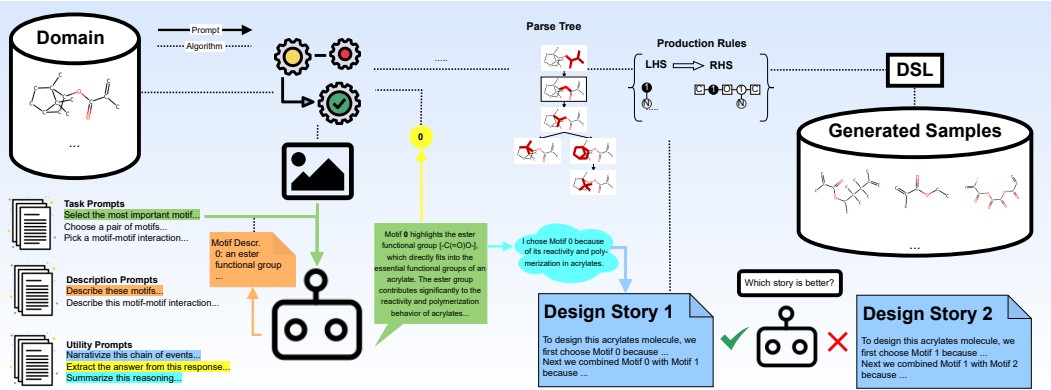

Figure 2: Our workflow takes as input a class-specific dataset and a collection of prompts (left); executes the tree decomposition algorithm with MMFM as a decision-making module (left middle) ; converts the parse tree into production rule set (left-right), resolving discrepancy across runs with a non-expert LLM judge; infers a DSL which can generate new class-specific samples (right).

### 3.1 PRELIMINARIES

**Molecular Clique Graph**. A base molecular hypergraph is a pair $H = (V_H, E_H)$, where $V_H$ (nodes) is a set of bonds, and $E_H$ (hyperedges) is a set of non-empty subsets of $V_H$. We follow prior work Kajino (2019); Guo et al. (2022b) and define $E_H :=$ $\{\{u, v\}$ if u, v share an atom $\} \bigcup \{\{u_i, 1 \leq i \leq k\}|\{u_i\}$ is a minimal ring $\}$. Given $H$, we obtain $G_H$, the graph of $H$, where two nodes $u, v$ sharing a common hyperedge in $E_H$ are connected. If we can construct a $G_C = (V_C, E_C)$ by extracting the maximal cliques ($V_C$) from $G_H$, and setting $E_C$ to be the clique pairs sharing a common node, we call $G_C$ the molecular clique graph and denote this operation as $CLIQUE(G_H) = G_C$. $G_C$ forms the building blocks for further operation. For each $c \in V_C$, we use $V_c$ to denote the clique nodes of $G_H$ within the clique $c$.

**Clique Tree Decomposition**. The clique tree, also known as junction tree, of $G_H$ is a tree $T$, each of whose nodes $\eta$ is labeled with a $V_\eta \subseteq V$ and $E_\eta \subseteq E$, such that the following properties hold: 1) For each $v$ in $G_H$, there is at least a vertex $\eta \in T$ such that $v \in V_\eta$. 2) For each hyperedge $e_i \in E$, there is exactly one node $\eta \in T$ such that $e \in E_\eta$ and $u \in e_i \rightarrow u \in V_\eta$. 3) For each $v \in G_H$, the set $\{\eta \in |T \mid v \in V_\eta\}$ is connected. The last property is the running intersection property and is relevant during the clique tree construction phase, as it needs to be checked after each step. The Junction Tree Algorithm achieves this by finding a subset $E_C' \subseteq E_C$, such that $(V_C, E_C')$ is a spanning tree of $G_C$. There is a theoretical guarantee that if $G_H$ is triangulated, there is always a valid tree decomposition. Choosing the best spanning edges $E_C'$ is somewhat of an art. There is the "optimal" clique tree, the one with minimal width := $\max(|V_\eta - 1|)$, but finding it is NP-hard. Instead, common heuristics like the maximum cardinality heuristic are used to find one close to minimal width.

**Hyperedge Replacement Grammar**. A hypergraph is a pair $H = (V_H, E_H)$ where $V_H$ is a set of nodes, and $E_H$ is a set of non-empty subsets of $V_H$, called hyperedges. A Hyperedge Replacement Grammar (HRG) is a tuple $(N, T, S, P)$ where: N are a set of non-terminal hyperedge labels in $\mathcal{N}$ T is a set of terminal hyperedge labels $S \in N$ is the starting non-terminal hyperedge with label 0 P is a set of production rules, each consisting of $A \in N$ (LHS) and R, a hypergraph with labeled hyperedges and —A— external nodes (RHS).

We adopt an automatic way to convert a clique tree into a HRG by interpreting the clique tree as a parse tree Aguinaga et al. (2018), where each intermediate node $V_\eta$ becomes the RHS of a production rule and its immediate parent and/or children are used to compute its non-terminal hyperedges and external nodes, as depicted in Fig. 3.

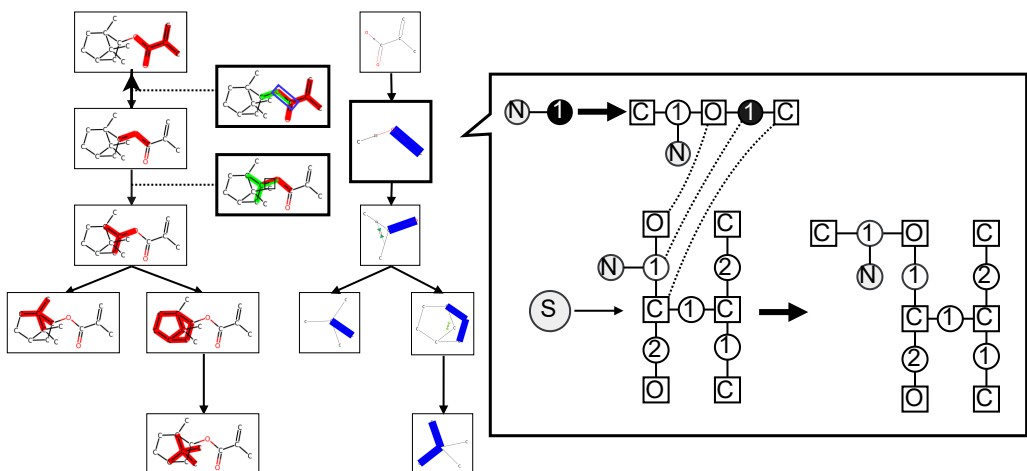

Figure 3: Conversion from clique tree to HRG production rules, an example rule application is shown for reconstructing the molecule parse tree

## 3.2 MMFM MODULES

For inducing a desirable DSL for molecular discovery, the gold standard is expert judgment. The essence of our approach is to modularize these exercises of judgment so an MMFM only needs to select amongst a finite set of choices in each module. These choices are captured by only two fundamental selections, which we now describe.

### 3.2.1 FUNDAMENTAL SELECTIONS

**Single Selection**. Given a set $S \subseteq V_C^{(t)}$, the MMFM is asked to select $s \in S$ or refrain from selection.

**Pair Selection**. Given a subset of pairs, $P \subseteq V_C^{(t)} \times V_C^{(t)}$, the MMFM is asked to select $p \in P$ or refrain from selection.

When the context is clear, we denote the raw responses $F_1(S^{(t)})$ and $F_2(P^{(t)})$. We use answer extraction utility prompts to obtain the answers. These selections map to triangulation, merging, cycle removal and root selection operations on $G_C^{(t)}$. We can execute the full tree decomposition of a molecular clique graph, $G_C^{(0)} \Rightarrow G_C^{(T)}$, using only these operations, driven by MMFM's selections. We will describe each operation $G_C^{(t)} \to G_C^{(t+1)}$, in detail, in the context of constructing the clique tree in Section 3.3.

### 3.2.2 PROMPTING SETUP

For each selection, we prompt the MMFM with rdkit rendered images and dynamical textual descriptions related to the current state of the decomposition ($G_C$), in addition to the static prompt, which includes some background on the domain and detailed task instructions.

**Rendering Images.** For single selection (root motif selection), we use the Python package rdkit for rendering the molecule and highlighting the bonds ($V_c$) of a single substructure ($c \in S^{(t)} \subseteq V_C^{(t)}$) into a cell. We use matplotlib.pyplot to enact a grid cell layout so all choices are shown together. For double selection where the number of choices are small (edge selection), we highlight each pair ($c_1, c_2 \in P^{(t)} \subseteq V_C^{(t)} \times V_C^{(t)}$) using different colors in the same cell. For double selection where the number of choices are large (merging cliques), we render each clique in a separate cell, just like with single selection, but the task instruction is to select a pair of cliques.

**Dynamic Textual Descriptions.** Motivated by the success of prompt-based learning techniques, we assist GPT's reasoning during selection tasks by plugging in isolated descriptions of each element of $S$ or $P$ into the task prompt, enabling multi-modal alignment. These are obtained by rendering each substructure (or pair of substructures) in isolation and asking GPT to describe those. An example of an isolated description is "Motif 5. Benzene - A six-membered aromatic ring entirely consisting of carbon atoms", whereas an in-context description is "Motif 5. A six-membered ring, similar to benzene, but includes distinct locations for double bonds from Motif 1."

**Rephrasing Prompts.** We then use format conversion prompts to convert GPT's sometimes elaborative answers into simple phrases that can be grammatically inserted into subsequent task prompts (example: "Motif 9. This motif is another carbocyclic structure, specifically a bicyclic system with carbon double bonds..." $\rightarrow$ "a bicyclic carbocyclic structure with carbon double bonds").

**Task Prompts.** These are the primary prompts for the workflow which instructs GPT to do the selection. We substitute rephrased dynamic descriptions of individual cliques (motifs) where appropriate into these templates and specifically instruct GPT to explain its reasoning. Example walkthroughs featuring all the task prompts are given in the Appendix.

**Answer Extraction Prompts.** We use low-level utility prompts for post-processing an answer prompt into a fixed format for regex extraction (example: "After extensive deliberation, the interaction between Motif 5 and Motif 7 seems weakest of the ones shown" $\rightarrow$ "5,7")

**Thought Collection Prompts.** We collect GPT's responses into summarized reasons for a particular selection, as they will be composed into a narrative (more in Section 3.4). For a particular selection at time $t$, let $COT(F_{j^{(t)}})$ be the prompt chaining composition to return a summarized reasoning over the selection. We denote the output as $COT^{(t)}$.

### 3.3 MMFM Guided Tree Decomposition Construction of Clique Graph

We initialize $G_H^{(0)}$ to the graph of the base molecular hypergraph. We extract the maximal cliques of $G_H^{(0)}$, thereby constructing $G_C^{(0)} \leftarrow CLIQUE(G_H^{(0)})$.

#### 3.3.1 Triangulate Clique Graph

We now triangulate $G_H^{(0)}$ to ensure the soundness of the junction tree algorithm. We adopt a chordality testing algorithm (Tarjan & Yannakakis, 1984) which iteratively detects pairs $(u, v) \in V_H \times V_H$ that would form chordless cycles of length $> 3$ if left unaddressed. At each iteration t that the algorithm returns a pair $(u, v)$ which must be connected via a chord, we set $P^{(t)} \rightarrow \{(c_1, c_2) \mid c_1 \in V_u \cap c_2 \in V_v\}$. Let $c^*_1, c^*_2 \leftarrow F_2(P^{(t)})$. We then merge $c^*_1, c^*_2$ by adding all edges, $E_H^{(t+1)} \leftarrow E_H^{(t)} \cup V_{c^*_1} \times V_{c^*_2}$. We update $G_C^{(t+1)} \leftarrow \text{CLIQUE}(G_H^{(t+1)})$. Let $G_C^{(T_1)}$ denote the clique graph once $G_H$ is triangulated. We proceed to the next phase.

#### 3.3.2 Merge Clique Nodes

We now would like to give the MMFM the option to further merge cliques that form more cohesive motifs, e.g. functional groups, in the context of the base molecule. Starting with $t = T_1$, we set $P^{(t)} \leftarrow E_C^{(t)}$. If $F_2(P^{(t)})$ does not return, we terminate and proceed to the next phase. Otherwise, at each iteration, we let $c^*_1, c^*_2 \leftarrow F_2(P^{(t)})$. We merge $c^*_1, c^*_2$ following the same operation steps as Step 2. Let $G_C^{(T_2)}$ denote the clique graph upon termination of this phase.

#### 3.3.3 Spanning Tree Edge Elimination

We now extract a spanning tree over $E_C^{(T_2)}$ using a top-down approach of detecting and eliminating cycles of $G_C^{(T_2)}$. We terminate and proceed to the next phase once there are no more cycles. Otherwise at each step t, let $c_1, c_2, \ldots, c_k, c_1$ be one such cycle. We set $P^{(t)} \leftarrow \{(c_i, c_{(i+1)\%k}) \mid \text{removing } c_i, c_{i+1} \text{ will not violate running intersection }, i = 1, 2, \ldots, k\}$. We then update $E_C^{(t+1)} \leftarrow E_C^{(t)} \setminus \{F_2(P^{(t)})\}$. Let $G_C^{(T_3)}$ denote the clique tree once all cycles have been removed.

### 3.3.4 ROOT MOTIF SELECTION

Lastly, we root $G_C^{(T_3)}$ at $F_1(V_C^{(T_3)})$. The final clique tree is $G_C^{(T)}$ ($T = T_3 + 1$). We obtain the multi-set of production rules using this decomposition, $\mathcal{P}(G_C^{(T)})$.

### 3.4 MMFM DRIVEN FMG LEARNING

Our MMFM-guided algorithm is inherently stochastic, as repeated runs may produce different decompositions. In the absence of human experts, it's difficult to judge how "good" the rules produced by each decomposition are. (Guo et al., 2022b) opts for learning the agent parameters via reinforcing distribution metrics of generated samples from the DSL (e.g. diversity, retrosynthesis score), but this way of overfitting to a task neglects the intrinsic qualities of the DSL. The key challenge is that given only the DSL, it's difficult to come up with the right metrics for its qualities. Our approach's built-in interpretability offers a new avenue to addressing this challenge. We repurpose the natural language artifacts (e.g. chain of thought, explanations) logged during our algorithm's execution as a proxy for the DSL's quality. With this point in mind, we adopt a simple yet effective learning procedure to optimize the FMG. We first perform K passes (i.e. independent runs of the algorithm) over the molecule $H$, producing decompositions $[G_{C_k}, k = 0, \ldots, K-1]$. Denoting $[COT_k^{(t)}, t = 0, \ldots, T-1]$ as the chain of thoughts for the k'th pass over molecule H, we combine it with knowledge of the timestep delimiters $T_1, T_2, T_3$ to compose a step-by-step story of how the molecule was decomposed. The resulting story becomes a proxy certification for the algorithm's correctness, and is further pitted against stories of discrepant decompositions for comparison by a non-expert LLM. Recent work (Khan et al., 2024) shows weaker LLMs can enhance stronger models via judging for persuasiveness while improving strong LLM's persuasiveness can even help weaker LLMs better identify the truth. Our FMG learning is optimizing for design stories that are persuasive to the non-expert, which can synergistically improve the judging quality. To optimize for persuasive design stories, we opt for a debate tournament. We pit discrepant runs (A and B) against each other in a debate, and ask the vanilla LLM to decide which story wins (A or B) on the basis of validity, soundness, and perceived depth of understanding. We adopt a Swiss tournament format, and use the logits of the first token in the response to assign outcomes of the matchup, similar to how (Khan et al., 2024) designed the preference model. We consolidate all outcomes using the Bradley-Terry Model (Bradley & Terry, 1952), a statistical model used for paired comparisons, where each debater's ability is inferred from the pairwise outcomes. We rank and order the participants $[0, 1, \ldots, K-1] \overset{\text{permute}}{\to} [r_1, r_2, \ldots, r_K]$ according to the outcomes of the tournament and define the "Top k" FMG as the HRG inferred by the production rule multi-set $\bigcup_{r \in \{r_1, \ldots, r_k\}} P(G_{C_r})$, where $\bigcup$ is the multiset union.

### 3.5 FMG INFERENCE AND STOCHASTIC SAMPLING FOR MOLECULAR GENERATION

So far, we have only considered the contribution to the HRG by decomposing a single molecule, H. In the domain-specific setting, we are given a small dataset of class-specific molecules (N ¡500), which we convert into our base molecular hypergraphs: $\mathcal{D} := \{H^{(i)} \mid 1 \le i \le N\}$. The DSL learning algorithm should adapt to $\mathcal{D}$ as a distribution, exposing parameters for inference. Similar to Aguinaga et al. (2018), we maintain a count for the number of times each rule is applied, aggregated across the top k runs for each $H^{(i)}$. During generation, the algorithm finds all applicable rules, and chooses one with probability proportional to its count. The derivation procedure for HRGs follows its common definition (Drewes et al., 1997). We adopt (Kajino, 2019)'s technique to ensure valid conversion from hypergraph to molecule.

## 4 RESULTS

We evaluate our method against other grammar-based and VAE methods, focusing on three main attributes of the generative model: **Synthesizability**, **Specificity** and **Coverage**. We evaluate on three small monomer datasets used by (Guo et al., 2022b) curated from literature, as well as two real-world datasets from the photovoltaic and toxicology domains used by (Sun et al., 2024). We use common unconditional generation metrics adopted by molecular generative models (Polykovskiy et al., 2020): **Valid/Unique/Novelty** (percentage of valid/unique/novel molecules) **Diversity** (average pairwise Tanimoto distance (Rogers & Hahn, 2010)) **Retro\* Score** (success rate of Retro\* model

Table 1: Results on Small Datasets Isocyanates (11), Acrylates (32) and Chain Extenders (11)

| Method | Unique | | | Div. | | | RS | | | Memb. | | |
|---|---|---|---|---|---|---|---|---|---|---|---|---|
| Train Data | 100% | 100% | 100% | 0.61 | 0.67 | 0.80 | 100% | 100% | 100% | 100% | 100% | 100% |
| JT-VAE | 5.8% | 0.5% | 2.3% | 0.72 | 0.29 | 0.62 | 5.5% | 4.9% | 2.2% | 66.5% | 48.64% | 79.6% |
| Hier-VAE | 99.6% | 99.7% | 99.8% | 0.83 | 0.83 | 0.83 | 1.85% | 3.04% | 2.69% | 0.05% | 0.82% | 43.6% |
| MHG | 75.9% | 86.8% | 87.4% | **0.88** | **0.89** | 0.90 | 2.97% | 36.8% | 50.6% | 12.1% | 0.93% | 41.2% |
| STONED | **100%** | 99.8% | 99.8% | 0.85 | 0.84 | **0.93** | 5.63% | 11.2% | 6.78% | 79.8% | 47.9% | 61.0% |
| DEG | **100%** | **100%** | **100%** | 0.86 | 0.86 | **0.93** | 27.2% | 43.9% | 67.5% | 96.3% | 69.6% | 93.5% |
| FMG | **100%** | **100%** | **100%** | 0.73 | 0.46 | 0.85 | **61.7%** | **93.0%** | **99.1%** | **99.6%** | **100%** | **99.8%** |

Table 2: Results on Medium Datasets HOPV (316) and PTC (348)

| Method | Unique | | Novelty | | Div. | | RS | | Memb. | |
|---|---|---|---|---|---|---|---|---|---|---|
| Train Data | 100% | 100% | N/A | N/A | 0.86 | 0.94 | 51% | 87% | 100% | 30% |
| JT-VAE | 11% | 8% | **100%** | 80% | 0.77 | 0.83 | **99%** | **96%** | **84%** | **27%** |
| Hier-VAE | **43%** | 20% | 96% | **85%** | **0.87** | 0.91 | 79% | 92% | 76% | 25% |
| Hier-VAE (expert) | 29% | **28%** | 92% | 75% | 0.86 | **0.93** | 84% | 90% | 82% | 17% |
| DEG | 98% | 88% | 99% | 87% | **0.93** | **0.95** | 19% | 38% | 46% | 27% |
| RW (expert) | **100%** | **100%** | **100%** | **100%** | 0.89 | 0.93 | 58% | 60% | **71%** | 22% |
| FMG | **100%** | **100%** | **100%** | 92% | **0.93** | 0.93 | **70%** | **78%** | 38% | **46%** |

(Chen et al., 2020) **Membership** (percentage of molecules belonging to the dataset's monomer class)[1].

We first observe in Tables 1 and that VAE methods struggle to generate unique molecules, suggesting they collapse in this extreme setting, consistent with findings by (Guo et al., 2022b; Sun et al., 2024). Hier-VAE fares better, as it incorporates inductive bias of larger substructures, but this comes at the expense of RS and Memb., suggesting an undesirable shift in distribution. The other two grammar-based methods do better on 3), but struggle across dimensions 2) and 3). Despite *optimizing* for RS and Div., DEG still falls short of FMG. The synthesizability scores are even more impressive knowing that we only prompted GPT to "highlight the primary functional groups of the molecule". FMG also achieves nearly 100% class membership in 1, suggesting FMG is sufficiently knowledgeable about these three chemical classes that it implicitly captures the constraint during its selections. This suggests domain-general FMs are already aligned with chemistry-specific desiderata like synthesizability and specificity, promoting the intrinsic quality of the DSL. However, FMG still leaves some to be desired across 3). Our investigation reveals the learning procedure is inclined towards forming cliques representing more complex substructures which are characteristic of the chemical class or known to be synthetically accessible. The applicability of a rule decreases as the RHS becomes more complex, and so the DSL's coverage decreases. We suspect the low diversity to be due to this phenomenon occurring in the extreme setting of having $\approx 30$ or less samples, as that creates fewer rules which are less applicable. We see, however, the diversity is far more reasonable for PTC and HOPV in Table 4, as the size of the dataset becomes larger. There, we still see VAE methods struggle similarly. The low uniqueness and novelty of the VAE baselines invalidates its seemingly high RS score, achieved by sampling smaller molecules. By contrast, FMG is one of only two methods who achieve 100% uniqueness (the other being RW with access to expert annotations) while tying for first and second on diversity for HOPV and PTC, respectively. Amongst grammar-based methods, FMG surpasses even RW on RS (by 12% and 18%), suggesting FMG is more amenable to synthesis considerations even for larger, more hand-engineered molecules. Though membership is not strictly defined for these two domains, FMG appears to do exceptionally well for PTC (halides) but poor for HOPV (thiophenes), which is surprising considering. As we see later in 5.2, $k$ imposes a sharp tradeoff between Memb. and {Div., RS}, though FMG is capable of achieving exceptional numbers for either/or.

---

[1] We generate 10000 for small datasets and 1000 for HOPV/PTC, use the same Retro parameters and adopt the same membership motifs as (Guo et al., 2022b; Sun et al., 2024).

## 5 ABLATIONS

### 5.1 HEURISTIC VS MMFM MODULES

Table 3: We ablate each MMFM module separately by replacing with a heuristic.

| Method | Novelty | | | Div. | | | RS | | | Memb. | | |
|---|---|---|---|---|---|---|---|---|---|---|---|---|
| FMG Avg | 99.96+-0.01 | 99.86 | 99.94+-0.00 | 0.79+-0.01 | 0.83+-0.00 | 0.81+-0.02 | 44.3+-3.4 | 87.4+-1.5 | 91.9+-3.8 | 60.14+-13.63 | 35.48+-4.02 | 28.30+-13.25 |
| FMG Union | 99.96 | 99.87 | 99.94 | 0.81 | 0.83 | 0.84 | 78.7 | 97.2 | 98.8 | 64.42 | 37.88 | 22.07 |
| FMG (-merge) Avg | 99.95+-0.00 | 99.88+-0.00 | 99.94+-0.00 | 0.74+-0.01 | 0.83+-0.00 | 0.85+-0.00 | 32.6+-5.7 | 91.0+-2.0 | 97.4+-0.8 | 95.75+-4.16 | 16.61+-0.78 | 15.48+-1.11 |
| FMG (-merge) Union | 99.95 | 99.88 | 99.94 | 0.76 | 0.83 | 0.85 | 39.7 | 90.3 | 96.4 | 93.74 | 16.40 | 14.44 |
| FMG (-edge) Avg | 99.96 | 99.87 | 99.95 | 0.76 | 0.82 | 0.77 | 57.9 | 93.5 | 99.9 | 45.81 | 37.44 | 38.56 |
| FMG (-edge) Union | 99.95 | 99.87 | 99.95 | 0.81 | 0.83 | 0.84 | 66.8 | 92.7 | 98.4 | 58.57 | 33.83 | 16.23 |
| FMG (-root) Avg | 99.96+-0.01 | 99.88+-0.00 | 99.94+-0.00 | 0.79+-0.03 | 0.85+-0.00 | 0.83+-0.02 | 49.1+-7.0 | 89.5+-2.6 | 91.9+-10.9 | 52.17+-12.13 | 22.90+-2.53 | 14.23+-6.39 |
| FMG (-root) Union | 99.97 | 99.86 | 99.94 | 0.82 | 0.85 | 0.86 | 54.9 | 87.0 | 96.2 | 47.01 | 22.18 | 14.84 |

We ablate each MMFM-assisted module to investigate how crucial each module is for bringing out the advantages of FMG. We ablate the merge module by directly passing $G_C^{(T_1)}$ to Step 3.3.3. We ablate the spanning tree module by adopting the common heuristic of the maximal spanning tree, where edge weights are assigned by cardinality of the intersection. We ablate the root module by picking a root clique at random. Since ablating an LLM module also breaks the overall design story, we only use the baseline "1-k" FMG (FMG Union, which combines all rules across K seeds). We set $K = 5$ and also report the average performance across 5 different runs. In Table 3, we see that removing any LLM component has negative implications for the results, albeit in different ways and differently for different datasets. When removing the merge step, the class-defining motifs for acrylates and chain extenders can no longer be formed during the decomposition, meaning they are less likely to be within the same clique and therefore appear in its entirety in the RHS of any rule. There is an exception for isocyanates, whose defining motif (N=C=O) has only 2 bonds and must be already part of a clique. For isocyanates, however, RS score drops significantly. It's known an amine (R-NH2) has to react with the phosgene (COCl2) to produce the isocyanate, so without the MMFM's knowledge, the synthetically accessible intermediate may not be formed, resulting in rules which are less amenable to synthetic considerations. When ablating the MMFM guided spanning tree construction, we see milder negative implications. Diversity, RS, and membership are all slightly worse, but there are no sharp drop offs. The maximal spanning tree heuristic is well-motivated from a theoretical point of view (Tarjan & Yannakakis, 1984), but its rule-based selection is less adaptable to domain-specific constraints like chemical reactivity and more rigid in modeling the interaction strength solely on the basis of neighborhood overlap. Meanwhile, an MMFM operating within the same framework is more flexible to capture these constraints, selectively breaking the rules when the context necessitates it.

### 5.2 ENSEMBLE OVER SEEDS

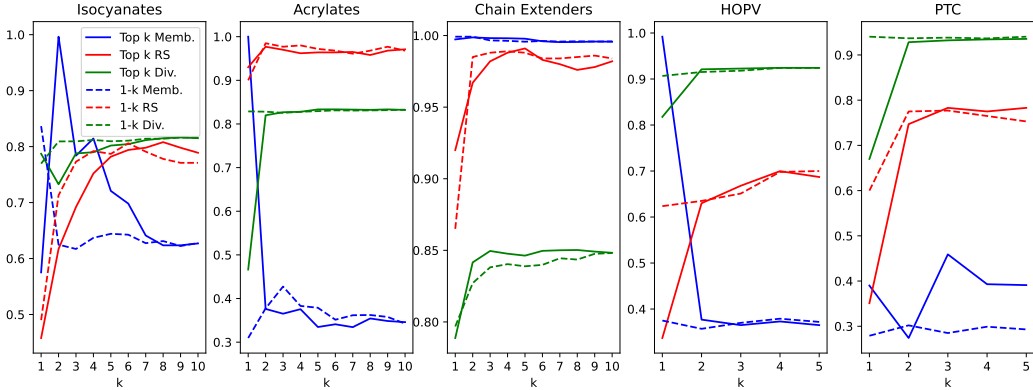

Figure 4: We vary $k$ from 1-10 (small dataset) and 1-5 (medium dataset) following the same settings as the main results.

We investigate the effect of the FMG learning in a more controlled setting. We set K=10 and host a Swiss style tournament with 4 rounds. We then study the performance of Top k FMG as $k$ increases. As a baseline, we compare with the "1-k" FMG, which is the HRG inferred by $\bigcup_{r=0}^{k-1} P(G_{C_r})$.

We find there are sharp tradeoffs in the generation metrics as k increases. We make several observations. First, it is easy to achieve near 100% membership for low values of k. This is because one of the points of comparison when evaluating two discrepant design stories being, "Which analysis better highlights the defining motif(s) of the acrylates chemical class?" We can deduce that 1) for each molecule, running for sufficient number of seeds always produces some decomposition that embeds the chemical class's defining motif within one of the rules, and 2) FMG is capable of ranking decompositions containing that property higher than those that do not. As a corollary, membership drops as k increases, as rules from sub-optimal decompositions are added to the DSL. Second, domain-specificity has some intrinsic tradeoff with synthesizability. Isocyanates are known to be tricky to synthesize due to unwanted side reactions. Choosing decompositions with design stories demonstrating a thorough understanding of the domain is more likely to overcomplicate the DSL from a synthesizability perspective. We also note some general trends as k increases. Diversity and RS seem to improve as more rule sets are combined. This is likely because a larger collection of "simple" rules, formed by alternative decompositions, enables more simple molecules to be generated, albeit at the cost of membership. Interestingly, there are no major differences between Top k and 1-k for RS and diversity, suggesting the learning procedure targets mainly class-specific considerations, remaining neutral to more general considerations.

## 6 DISCUSSION

We introduce a MMFM guided DSL induction algorithm and show a specific application for molecular discovery. We introduce a general recipe for integrating MMFM's knowledge and reasoning capabilities into a sound DSL induction framework, formulating the MMFM's task as a sequence of selections. We introduce innovative techniques in prompting, rendering and evaluation to prime the MMFM to reason like a domain expert over molecular graphs. Our evaluation on molecular generation benchmarks shows expert-like ability to decompose a molecule while indirectly capturing human preferences for specificity and synthesizability. Most importantly, our entire method is inviting to the end user, who can control the prompts, edit the selections or ideate off the MMFM's reasonings. Our simple learning and inference framework is simple, while laying the foundation for more sophisticated techniques for closed-loop optimization which can be the avenue for future research.

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
