## A CASE STUDIES, TASK PROMPTS, EXPERT ANALYSES, AND DESIGN STORY

In the Appendix, we do turn-by-turn walkthroughs of representative molecules from every dataset. We show logs of every task-related (prompt, response) pair used in our algorithm. For closure of our work, we invite a real chemist to comment on the algorithm's logs for each dataset and comment on a) the difficulty of the prompt and b) GPT's answers. We preface each dataset (subsection) with a brief overview of the domain and rationale for choosing the specific molecule. For the interested reader, we also include a sample design story used in learning the FMG. As the unrefined stories are heavy in jargon and quite elaborative, we only include one example. We choose a HOPV molecule as they feature the richest stories. We include the full text under HOPV's Section A.2.

In Section A.6, we leave with concluding remarks on the performance of GPT-4o and speculate on its potential.

### A.1 CASE STUDY: PREDICTIVE TOXICOLOGY CHALLENGE (PTC)

First, we establish some formatting conventions that we use for PTC and HOPV:

- We log the task prompt, the corresponding response and the expert's comments for each call to GPT-4o, subdivided by header cells labeling what's in the following cell.
- We omit utility prompts like extracting integer answers and instead focus on the core selection tasks done by GPT.
- Dynamic substitutions are quoted in "'...'".
- Pre-written background specific to the domain is *italicized*.

For PTC, since it often contain halides and quaternary ammonium groups, which assist in transferring reactants between different phases in a reaction, the selected molecule picked not only have multiple symmetrical halide component but are also comprehensive enough for GPT to studying the rest of functional motif's relative impact to the entire molecule, making it an ideal candidate for studying phase-transfer catalysis in synthetic chemistry and investigating toxicity, given that some PTCs are known carcinogens.

Figure 5: Input molecule from PTC

| Prompt |
| --- |
| I want you to think like a chemist performing a detailed analysis of the chemical composition of a toxic compound through its constituent motifs. I will highlight for you "'24'" of the substructures of a molecule. They are numbered one-by-one from Motif 0 to Motif "'23'", inclusive. I want you to explain, concisely, what each numbered motif is. Make sure to start from Motif 0 and go in order of the numbering. MAKE SURE you describe EVERY MOTIF! |

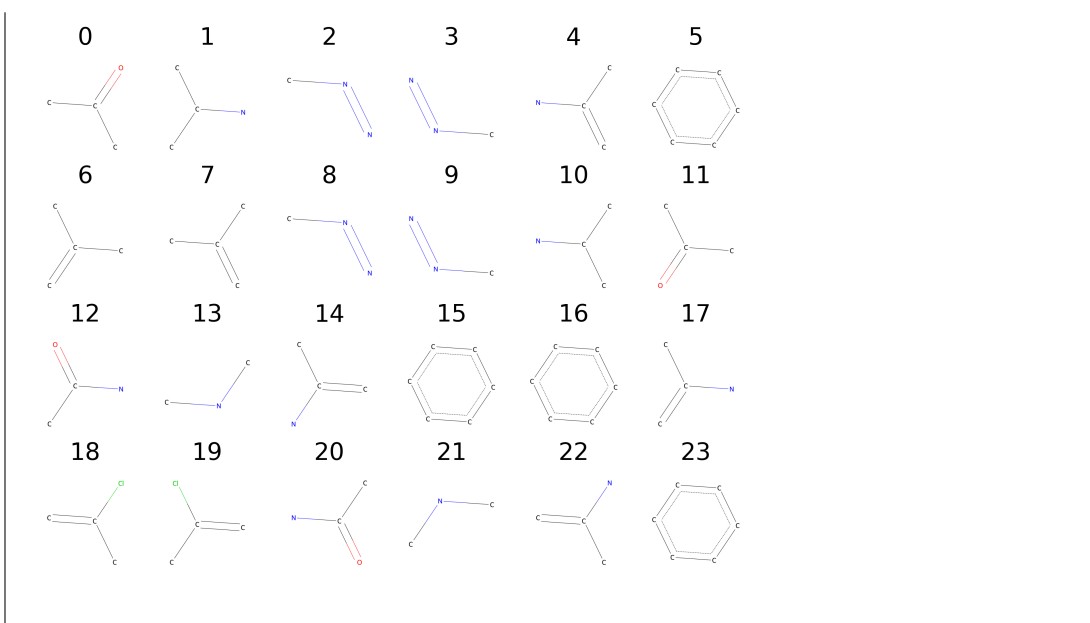

Response

Certainly! Here is a concise description of each motif:
**Motif 0:** A carbonyl group (C=O) attached to a carbon chain.
**Motif 1:** A nitrile group ($C \equiv N$) attached to a tertiary carbon.
**Motif 2:** A di-substituted carbon chain with two adjacent nitrile groups (N=C-C=C-N).
**Motif 3:** Similar to Motif 2 but with an additional carbon on one end.
**Motif 4:** An imine group (C=N) connected to a vinylic carbon chain (C=C).
**Motif 5:** Benzene ring (C6H6) - a six-membered aromatic carbon ring.
**Motif 6:** A singly unsaturated carbon chain.
**Motif 7:** A doubly unsaturated carbon chain.
**Motif 8:** Di-substituted carbon chain with a nitrile group (N=C) on one end and another nitrile ($N \equiv C$) in the middle.
**Motif 9:** Similar to Motif 8 but with different placement of carbons.
**Motif 10:** An amino group (NH2) attached to a C=N group (imino).
**Motif 11:** A ketone (C=O) with adjacent carbons.
**Motif 12:** A cyano-ketone ($C \equiv N$ and C=O) attached to adjacent carbons.
**Motif 13:** An alkyl cyanide ($R - C \equiv N$) where R is an alkyl chain.
**Motif 14:** An amino group (NH2) connected to a nitrile ($C \equiv N$).
**Motif 15:** A benzene ring (C6H6) - slightly different representation of Motif 5.
**Motif 16:** Another representation of a benzene ring (C6H6).
**Motif 17:** A nitrile group ($C \equiv N$) attached to a doubly unsaturated carbon chain.
**Motif 18:** An alkyl halide (carbon chain with a chlorine, C-Cl).
**Motif 19:** Similar to Motif 18 with a different number of carbons.
**Motif 20:** A doubly canned ketone (C=O) group attached to an imine (C=N) group.
**Motif 21:** An imine group (C=N) attached to an ethyl group (C2H5).
**Motif 22:** A doubly bonded carbon chain similar to an alkene.
**Motif 23:** Another benzene ring structure.
This list consists of a variety of functional groups and structural motifs commonly found in organic chemistry, ranging from aromatic rings to carbon-nitrogen triple bonds, and functional groups such as ketones and amines.

Comments by Expert

correctly answered and described all 0-23 motifs

Prompt

I want you to think like a chemist performing a detailed analysis of the chemical composition of a toxic compound through its constituent motifs. I will highlight for you some of the distinctive fragments of a molecule. They are numbered from 0 and individually highlighted in GREEN. Focus ONLY on the substructure highlighted in green within each cell. Here is the descriptions for each substructure provided by an expert:

"'Certainly! Here is a concise description of each motif:
**Motif 0:** A carbonyl group (C=O) attached to a carbon chain.
**Motif 1:** A nitrile group ($C \equiv N$) attached to a tertiary carbon.
**Motif 2:** A di-substituted carbon chain with two adjacent nitrile groups (N=C-C=C-N).
...
**Motif 23:** Another benzene ring structure.
This list consists of a variety of functional groups and structural motifs commonly found in organic chemistry, ranging from aromatic rings to carbon-nitrogen triple bonds, and functional groups such as ketones and amines."'

I want you to tell me if any two of them should be combined together to form a more meaningful substructure. *This molecule belongs to a collection of molecules characterized by distinct functional groups known for their carcinogenic properties or liver toxicity. These groups comprise a rich variety of elements such as halides, alkylating agents, epoxides, and furan rings.* Your task is to highlight the primary functional groups of the molecule. Output a single pair of numbers if you think those two fragments should be combined, and a brief explanation why. If no such pairs exist, don't output anything.

### Response

**Pair: 8 and 9**
**Explanation:** Motifs 8 and 9 are both di-substituted carbon chains with nitrile groups in different positions. Combining these two fragments could provide a more comprehensive understanding of the arrangement and connectivity of nitrile groups along the carbon chain, which is critical in analyzing the structure and reactivity of the molecule, especially in the context of toxic compounds where such functional groups can have significant implications for toxicity.

### Comments by Expert

medium, PTC molecules are more complex, so more than one motifs could be combined together

### Prompt

I want you to think like a chemist performing a detailed analysis of the chemical composition of a *toxic compound* through its constituent motifs. I will highlight for you "'23'" of the substructures of a molecule. They are numbered one-by-one from Motif 0 to Motif "'22'", inclusive. I want you to explain, concisely, what each numbered motif is. Make sure to start from Motif 0 and go in order of the numbering. MAKE SURE you describe EVERY MOTIF!

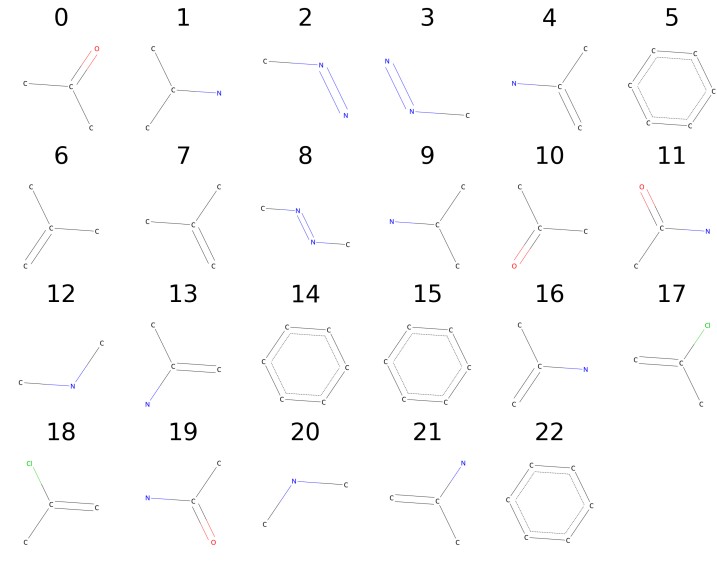

Response

Certainly! Here are the chemical motifs numbered from 0 to 22:

**Motif 0**: Acetone (propan-2-one) - A simple ketone with the formula $(CH_3)_2CO$.

**Motif 1**: Acetonitrile - A nitrile compound where the CN group is attached to a methyl group.

**Motif 2**: Fumaroid nitrile - A dinitrile with both nitrile groups on opposite sides.

**Motif 3**: Cis-maleonitrile - A dinitrile with both nitrile groups on the same side.

**Motif 4**: Acrylonitrile - A nitrile compound with a vinyl group attached to the CN.

**Motif 5**: Benzene - An aromatic hydrocarbon with a six-membered carbon ring and alternating double bonds (represented here with a hexagon).

**Motif 6**: Propene (propylene) - An alkene with the chemical formula $C_3H_6$ containing a double bond between two carbon atoms.

**Motif 7**: 2-Butene - An alkene which has a double bond between the second and third carbons of a four-carbon chain.

**Motif 8**: Butanedinitrile - A nitrile with a CN group at each end of a butane backbone.

**Motif 9**: Acrylonitrile derivative - Similar to acrylonitrile with an additional methyl group.

**Motif 10**: Methacrolein - An aldehyde with a carbon-carbon double bond.

**Motif 11**: Ethyl cyanide (propanenitrile) - A nitrile with a three-carbon chain.

**Motif 12**: Propan-1-amine - An amine with a three-carbon chain.

**Motif 13**: Tert-butyl - A bulky, branched alkyl group with three methyl groups attached to a single carbon atom.

**Motif 14**: Benzene (repetition of Motif 5) - Another illustration of benzene.

**Motif 15**: Benzene (repetition of Motif 5 and 14) - Yet another illustration of benzene.

**Motif 16**: Isobutyraldehyde - An aldehyde with a branched three-carbon chain.

**Motif 17**: Chloroacetylene - An alkyne with chlorine attached to a carbon-carbon triple bond.

**Motif 18**: 1,1-Dichloroethane - An alkane with two chlorine atoms attached to the same carbon atom.

**Motif 19**: 2-Methylpropanoylnitrile - A nitrile with a methyl group attached to a carbon atom that is part of the nitrile structure.

**Motif 20**: Benzene (repetition of Motifs 5, 14, and 15) - Another repetition of benzene.

**Motif 21**: Acetonitrile again (as in Motif 1) - Another nitrile example.

**Motif 22**: Benzene - final repetition of the benzene structure.

This detailed description covers the unique structural motifs presented in the molecule, from acetone to benzene, and other functional groups like nitriles and alkenes.

Prompt

I want you to think like a chemist performing a detailed analysis of the chemical composition of a *toxic compound* through its constituent motifs. I will highlight for you some of the distinctive fragments of a molecule. They are numbered from 0 and individually highlighted in GREEN. Focus ONLY on the substructure highlighted in green within each cell. Here is the descriptions for each substructure provided by an expert:

"'Certainly! Here are the chemical motifs numbered from 0 to 22:

**Motif 0**: Acetone (propan-2-one) - A simple ketone with the formula $(CH_3)_2CO$.

**Motif 1**: Acetonitrile - A nitrile compound where the CN group is attached to a methyl group.

... **Motif 22**: Benzene - final repetition of the benzene structure.

This detailed description covers the unique structural motifs presented in the molecule, from acetone to benzene, and other functional groups like nitriles and alkenes.'"

I want you to tell me if any two of them should be combined together to form a more meaningful substructure. *This molecule belongs to a collection of molecules characterized by distinct functional groups known for their carcinogenic properties or liver toxicity. These groups comprise a rich variety of elements such as halides, alkylating agents, epoxides, and furan rings.* Your task is to highlight the primary functional groups of the molecule. Output a single pair of numbers if you think those two fragments should be combined, and a brief explanation why. If no such pairs exist, don't output anything.

### Response

**Pair: 2, 3**

**Explanation:** Motif 2 and Motif 3 both depict dinitriles, with variations in their positioning (fumaroid vs. cis-maleonitrile, respectively). Combining these motifs could result in a more complete representation of a molecule with multiple nitrile groups, which are significant regarding potential toxic and carcinogenic properties. This merged substructure can serve as a more insightful representation of the functional group within the overall toxic compound.

### Comments by Expert

This is partially correct, but combining 17 and 18 which contains the halide groups are also important

### Prompt

I want you to think like a chemist performing a detailed analysis of the chemical composition of a *toxic compound* through its constituent motifs. I will highlight for you some of the distinctive substructures of a *toxic compound*. They are numbered from 0. Here are the textual descriptions of each motif:

"'Certainly! Here are the chemical motifs numbered from 0 to 22:

**Motif 0**: Acetone (propan-2-one) - A simple ketone with the formula $(CH_3)_2CO$.

**Motif 1**: Acetonitrile - A nitrile compound where the CN group is attached to a methyl group.

...

**Motif 22**: Benzene - final repetition of the benzene structure.

This detailed description covers the unique structural motifs presented in the molecule, from acetone to benzene, and other functional groups like nitriles and alkenes.'"

I want you to pick only ONE of these as the root motif most essential to its chemical profile. It should be the single most important motif the rest of the *toxic compound* was built around. *This molecule belongs to a collection of molecules characterized by distinct functional groups known for their carcinogenic properties or liver toxicity. These groups comprise a rich variety of elements, most notably halides, so your selected root motif MUST contain a halide group.* If there are multiple such motifs, or one doesn't clearly stand out, just pick one of them. Give your answer as a single number. Explain your reasoning carefully.

| Comments by Expert |
| --- |
| Medium, pick only one of the most important motif can cause somewhat different thoughts between language models and chemistry experts |

| Response |
| --- |

Upon reviewing the chemical motifs, the selection criteria specify that the root motif should contain a halide group, which is characteristic of the molecule's toxic properties.

Examining through the motifs, certain ones include halides:

- **Motif 0**: Acetone does not have a halide.
- **Motif 1**: Acetonitrile does not have a halide.
- **Motif 2**: Fumaroid nitrile does not have a halide.
- **Motif 3**: Cis-maleonitrile does not have a halide.
- **Motif 4**: Acrylonitrile does not have a halide.
- **Motif 5**: Benzene does not have a halide.
- **Motif 6**: Propene does not have a halide.
- **Motif 7**: 2-Butene does not have a halide.
- **Motif 8**: Butanedinitrile does not have a halide.
- **Motif 9**: Acrylonitrile derivative does not have a halide.
- **Motif 10**: Methacrolein does not have a halide.
- **Motif 11**: Ethyl cyanide does not have a halide.
- **Motif 12**: Propan-1-amine does not have a halide.
- **Motif 13**: Tert-butyl does not have a halide.
- **Motif 14**: Benzene repetition does not have a halide.
- **Motif 15**: Benzene repetition does not have a halide.
- **Motif 16**: Isobutyraldehyde does not have a halide.
- **Motif 17**: Chloroacetylene **contains a halide**.
- **Motif 18**: 1,1-Dichloroethane **contains a halide**.
- **Motif 19**: 2-Methylpropanoylnitrile does not have a halide.
- **Motif 20**: Benzene repetition does not have a halide.
- **Motif 21**: Acetonitrile repetition does not have a halide.
- **Motif 22**: Benzene repetition does not have a halide.

From the motifs listed, **Motif 17 (Chloroacetylene)** and **Motif 18 (1,1-Dichloroethane)** contain halide groups. Both of these could potentially serve as the root due to their inclusion of a chlorine atom, which is a halide.

Given the need to select one motif, **Motif 18** (1,1-Dichloroethane) stands out slightly more due to the presence of two chlorine atoms, which could contribute to heightened toxicity and distinct chemical properties of the toxic compound.

Thus, the most essential root motif most likely to be built around is **Motif 18**.

---

Response

---

GPT describe both motif 17 and 18, as they both contains the halide function groups, and choose motif 18, which is correct

---

Prompt

---

I want you to perform a simple post-processing step of the following response:

'''Certainly! Here are the chemical motifs numbered from 0 to 22:

**Motif 0**: Acetone (propan-2-one) - A simple ketone with the formula $(CH_3)_2CO$.

**Motif 1**: Acetonitrile - A nitrile compound where the CN group is attached to a methyl group.

...

**Motif 22**: Benzene - final repetition of the benzene structure.

This detailed description covers the unique structural motifs presented in the molecule, from acetone to benzene, and other functional groups like nitriles and alkenes.''' The input is a response from another language agent. It describes motifs numbered from Motif 0 to '''Motif 22''', inclusive! I want you to rephrase each motif description by filling in X within the following sentence template: This motif is X Be sure to condense the description and output a single PHRASE such that the sentence template is grammatically correct. Don't capitalize the first letter, since your answer should just be a phrase. Output your rephrasing for each motif on a SEPARATE line, using only a new line for delimiting different motifs. Don't output anything else. MAKE SURE you do it for EVERY MOTIF!

---

Comments by Expert

---

Easy prompt

Response

an acetone (propan-2-one)
an acetonitrile
a fumaroid nitrile
a cis-maleonitrile
an acrylonitrile
a benzene
a propene (propylene)
a 2-butene
a butanedinitrile
an acrylonitrile derivative
a methacrolein
an ethyl cyanide (propanenitrile)
a propan-1-amine
a tert-butyl
a benzene
a benzene
an isobutyraldehyde
a chloroacetylene
a 1,1-dichloroethane
a 2-methylpropanoylnitrile
a benzene
an acetonitrile
a benzene

Comments by Expert

it summarizes all functional groups correctly

Prompt

I want you to think like a chemist performing a detailed analysis of the chemical composition of a molecule through its constituent motifs. This requires two steps: 1) analyzing the individual motifs and 2) analyzing the pairwise interactions of motifs. The first step is already done. The second step is where I need your help. I will highlight for you different motif interactions within the same molecule. These interactions are numbered one-by-one, beginning with 0. Here are the textual descriptions of each motif interaction pair.
Interaction "'0'" features "'a benzene'" and "'an isobutyraldehyde'".
Interaction "'1'" features "'an isobutyraldehyde'" and "'a chloroacetylene'".
Interaction "'2'" features "'a chloroacetylene'" and "'a benzene'".
I want you to tell me which one of these is MOST important, and which one of these is LEAST important. Output one number identifying the MOST important, and give a brief explanation. Output one number identifying the LEAST important, and a brief explanation why.

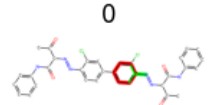 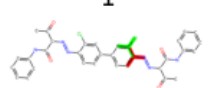

Comments by Expert

Hard, it requires abundant chemistry empirical experiences in organic synthesis, chemical compounds ' stability and etc

Response

To determine which interaction is the most and least important, it's useful to consider the chemical properties and potential intermolecular forces involved in these interactions.
### Interaction Details: 1. **Interaction 0**: Benzene and Isobutyraldehyde
2. **Interaction 1**: Isobutyraldehyde and Chloroacetylene
3. **Interaction 2**: Chloroacetylene and Benzene
### Analysis: - **Benzene**: Benzene rings often participate in $\pi - \pi$ stacking interactions, increasing molecular stability and electronic distribution. - **Isobutyraldehyde**: This functional group has a polar aldehyde component that can form hydrogen bonds and dipole-dipole interactions. - **Chloroacetylene**: This highly reactive group contains both an acidic hydrogen (from the acetylene) and a chloro substituent which may influence its reactivity and interaction propensity.
### Pairwise Interactions: - **Interaction 0 (Benzene and Isobutyraldehyde)**: This interaction may involve weak $\pi - \pi$ interactions from benzene and dipole-dipole attractions from the aldehyde group.
- **Interaction 1 (Isobutyraldehyde and Chloroacetylene)**: The aldehyde offers dipole-dipole interactions while the chloroacetylene provides potential for bonding interactions based on its triple bond and electronegative chloro group.
- **Interaction 2 (Chloroacetylene and Benzene)**: This interaction might involve $\pi - \pi*$ interactions from benzene and possible dipole-induced interactions from chloroacetylene.
### Most Important: **Interaction 1 (Isobutyraldehyde and Chloroacetylene)** Explanation: This interaction involves strong dipole-dipole interactions from the polar aldehyde group and the highly reactive chloroacetylene. The triple bond and the chlorine atom enhance the electrophilic character, making this interaction highly influential in the molecule's reactivity and stability.
### Least Important: **Interaction 0 (Benzene and Isobutyraldehyde)** Explanation: While benzene can participate in $\pi - \pi$ interactions, these non-covalent interactions are generally weaker compared to strong dipole-dipole or potential covalent interactions found in other pairs. The electrostatic attraction between benzene's $\pi-$ electrons and the aldehyde may not contribute as significantly to the molecule's overall properties.
### Summary: - **Most Important: Interaction 1** - **Least Important: Interaction 0**

---

Comments by Expert

---

This is fully correct, interaction 1 ( benzene and a 1,1-dichloroethane ) should be most important and 0 (acrylonitrile and a benzene) should be the least in this case.

---

Prompt

---

I want you to think like a chemist performing a detailed analysis of the chemical composition of a molecule through its constituent motifs. This requires two steps: 1) analyzing the individual motifs and 2) analyzing the pairwise interactions of motifs. The first step is already done. The second step is where I need your help. I will highlight for you different motif interactions within the same molecule. These interactions are numbered one-by-one, beginning with 0. Here are the textual descriptions of each motif interaction pair.
Interaction "'0'" features "'an acrylonitrile'" and "'a benzene'".
Interaction "'1'" features "'a benzene'" and a "'1,1-dichloroethane'".
Interaction "'2'" features "'a 1,1-dichloroethane'" and "'an acrylonitrile'".
I want you to tell me which one of these is MOST important, and which one of these is LEAST important. Output one number identifying the MOST important, and give a brief explanation. Output one number identifying the LEAST important, and a brief explanation why.

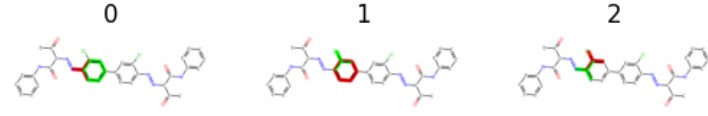

---

Response

---

**Most Important: Interaction 1**
Explanation: Interaction 1 features a benzene and a 1,1-dichloroethane. Benzene is a highly stable aromatic compound, and its interactions are often significant due to its resonance stability and the potential for $\pi-\pi$ stacking interactions. 1,1-Dichloroethane can interact through dipole-dipole interactions due to the presence of electronegative chlorine atoms. The combination of these two motifs could provide strong intermolecular or intramolecular interactions, impacting the overall stability and properties of the molecule significantly.
**Least Important: Interaction 0**
Explanation: Interaction 0 features an acrylonitrile and a benzene. Acrylonitrile, while reactive due to its nitrile group, may not interact as strongly with benzene compared to the involvement of more polar or reactive groups like the dichloroethane. The interaction with benzene here would primarily be through weaker van der Waals forces and does not have the significant dipole-dipole interactions present in Interaction 1.

## A.2   CASE STUDY: HARVARD ORGANIC PHOTOVOLTAIC DATASET (HOPV)

For HOPV, the selected molecules include several of the most significant functional motifs that in contributing to molecules' photovoltaic properties. These functional groups include the classical thiolphene unit, thiazole ring, pyrimidinedione ring and dithieno[3,2-b:2',3'-d]thiophene (DTT), it is a really classical, comprehensive and representative HOPV molecule.

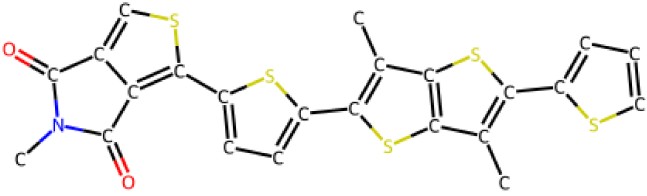

Figure 6: Input molecule from HOPV

Prompt

I want you to think like a chemist performing a detailed analysis of the chemical composition of a *photovoltaic molecule* through its constituent motifs. I will highlight for you 11 of the substructures of a molecule. They are numbered one-by-one from Motif 0 to Motif "'10'", inclusive. I want you to explain, concisely, what each numbered motif is. Make sure to start from Motif 0 and go in order of the numbering. MAKE SURE you describe EVERY MOTIF!

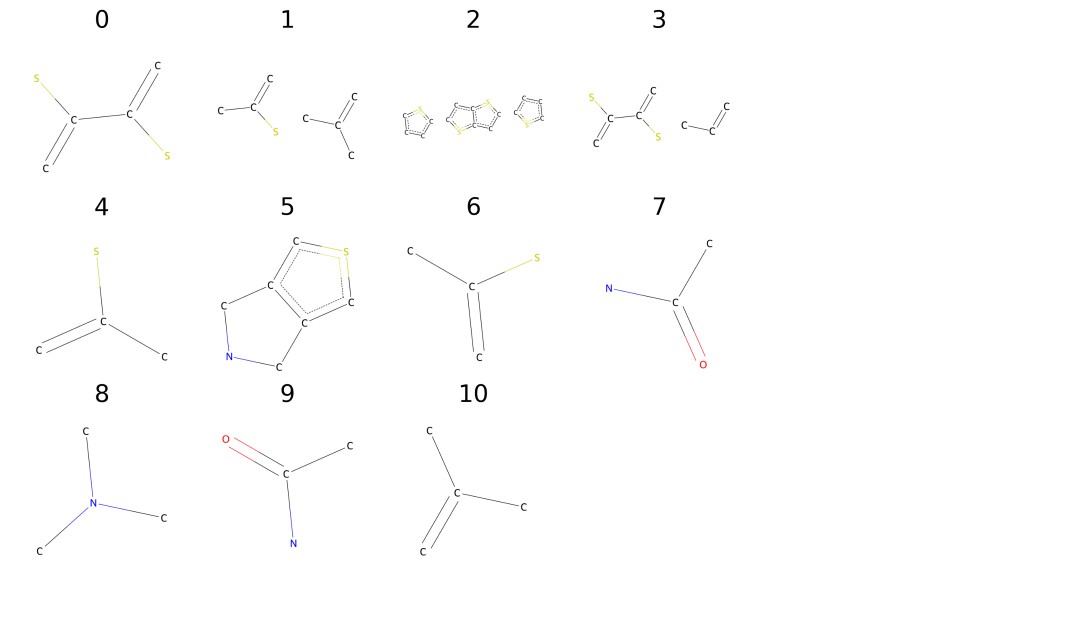

Response

Here's the detailed analysis of each motif in the given photovoltaic molecule, broken down:

1. **Motif 0**: - **Structure**: It is a 1,3-dithiol-2-one structural motif. It features a ketone (carbonyl group) with a five-membered ring containing two sulfur atoms adjacent to the ketone group.

2. **Motif 1**: - **Structure**: This motif consists of two distinct substructures: 1. A five-membered aromatic ring resembling thiophene, composed of a sulfur atom (yellow) and four carbon atoms. 2. A simple three-carbon chain with a double bond, ending in a ketone group (carbonyl group).

3. **Motif 2**: - **Structure**: This motif features a set of three different heterocyclic aromatic rings: 1. A thiophene ring (five-membered ring with sulfur). 2. A furan ring (five-membered ring with an oxygen atom). 3. A pyrrole ring (five-membered ring with a nitrogen atom).

4. **Motif 3**: - **Structure**: This is similar to Motif 0 but includes an additional carbonyl group (C=O) branching off the central carbon in the five-membered ring structure.

5. **Motif 4**: - **Structure**: This motif resembles a stylized thioacetone structure. It consists of a three-carbon chain with a sulfur atom connected to the central carbon and two other carbon atoms attached via single bonds.

6. **Motif 5**: - **Structure**: This motif features an indole-like structure. It is a bicyclic system with a benzene ring fused to a five-membered ring containing a nitrogen atom and a sulfur atom, resembling benzothiophene.

7. **Motif 6**: - **Structure**: This is a thioketo structure. It consists of a three-carbon chain with a sulfur atom double-bonded to the central carbon and a carbon attached via a double bond.

8. **Motif 7**: - **Structure**: This motif represents a nitrile ketone structure. It is a three-carbon chain where one carbon is triple-bonded to nitrogen (nitrile group) and double-bonded to oxygen (carbonyl group).

9. **Motif 8**: - **Structure**: This structure is a simple nitrile. It consists of a three-carbon chain with a nitrogen atom triple-bonded to the central carbon.

10. **Motif 9**: - **Structure**: This motif resembles acetyl cyanide. It is a three-carbon chain with a ketone group (carbonyl group) and a cyanide group (triple-bonded nitrogen).

11. **Motif 10**: - **Structure**: This is an isoprene structure. It features a four-carbon chain with two double bonds present, making up a simple conjugated diene.

These motifs collectively illustrate the diversity of functional groups and substructures possibly present in the complex photovoltaic molecule. Each motif brings distinct electronic and structural properties to the table, contributing to the molecule's overall function in photovoltaic applications.

---

Comments by Expert

---

Easy prompt with clear instructions to follow GPT's answer: correctly describe the name and functionality of all 11 motifs

---

Prompt

I want you to think like a chemist performing a detailed analysis of the chemical composition of a molecule through its constituent motifs. I will highlight for you some of the distinctive fragments of a molecule. They are numbered from 0 and individually highlighted in GREEN. Focus ONLY on the substructure highlighted in green within each cell. Here is the descriptions for each substructure provided by an expert:

'''Here's the detailed analysis of each motif in the given photovoltaic molecule, broken down:
1. **Motif 0**: - **Structure**: It is a 1,3-dithiol-2-one structural motif. It features a ketone (carbonyl group) with a five-membered ring containing two sulfur atoms adjacent to the ketone group.
2. **Motif 1**: - **Structure**: This motif consists of two distinct substructures: 1. A five-membered aromatic ring resembling thiophene, composed of a sulfur atom (yellow) and four carbon atoms. 2. A simple three-carbon chain with a double bond, ending in a ketone group (carbonyl group). ... '''

I want you to tell me if any two of them should be combined together to form a more meaningful substructure. This molecule belongs to a collection of molecules used to aid the design of organic solar cells. These molecules comprise of functional groups and complex ring structures that contribute to optimized photovoltaic efficiency, utilizing factors like Pi-orbital electron delocalization or electrondonating groups/structures increasing steric hindrance. Your task is to highlight the primary functional groups of the molecule. Output a single pair of numbers if you think those two fragments should be combined, and a brief explanation why. If no such pairs exist, don't output anything.

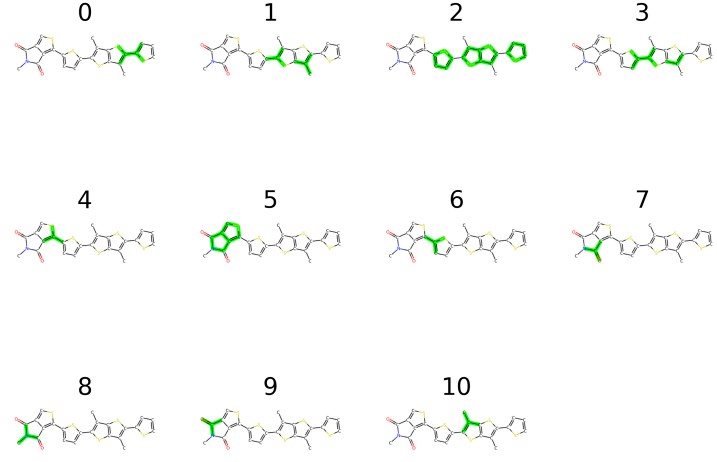

Response

No two fragments should be combined together to form a more meaningful substructure from the given motifs. Each motif represents distinct functional groups or structural elements that can individually contribute to the photovoltaic properties of the molecule.

Comments by Expert

Easy prompt, useful and specific instructions for GPT such as hint to utilize factors like electron delocalization, etc. GpT's answer: No two fragments should be combined together to form a more meaningful substructure from the given motifs. Each motif represents distinct functional groups or structural elements that can individually contribute to the photovoltaic properties of the molecule. This is correct, motif 1 provide the most distinct function to the molecule's optoelectronic properties, however, none of them form no meaningful substructure. They were injected before due to the ease of synthesis.

Prompt

I want you to think like a chemist performing a detailed analysis of the chemical composition of a *photovoltaic molecule* through its constituent motifs. I will highlight for you some of the distinctive substructures of a *photovoltaic molecule*. They are numbered from 0. Here are the textual descriptions of each motif:

"'Here's the detailed analysis of each motif in the given *photovoltaic molecule*, broken down:

1. **Motif 0**: - **Structure**: It is a 1,3-dithiol-2-one structural motif. It features a ketone (carbonyl group) with a five-membered ring containing two sulfur atoms adjacent to the ketone group.

2. **Motif 1**: - **Structure**: This motif consists of two distinct substructures: 1. A five-membered aromatic ring resembling thiophene, composed of a sulfur atom (yellow) and four carbon atoms. 2. A simple three-carbon chain with a double bond, ending in a ketone group (carbonyl group).

... ,"

I want you to pick only ONE of these as the root motif most essential to its chemical profile. It should be the single most important motif the rest of the molecule was built around. *This molecule belongs to a collection of molecules used to aid the design of organic solar cells. These molecules comprise of functional groups and complex ring structures that contribute to optimized photovoltaic efficiency, utilizing thiophene-based groups for enhanced Pi-orbital electron delocalization or electrondonating groups/structures increasing steric hindrance. Your selected root motif MUST contain the thiophene group.* If there are multiple such motifs, or one doesn't clearly stand out, just pick one of them. Give your answer as a single number. Explain your reasoning carefully.

---

Response

---

The root motif most essential to the chemical profile of the photovoltaic molecule, which contains the thiophene group and contributes significantly to the molecule's function in photovoltaic applications, is **Motif 1**.

**Reasoning:** - **Thiophene Presence**: Motif 1 contains a distinct thiophene unit, which is critical for enhanced $\pi$-orbital electron delocalization. Thiophenes are well-known for their role in organic electronics and photovoltaics due to their ability to facilitate charge transport through $\pi$-conjugation. - **Substructure Complexity**: Motif 1 consists of both a thiophene ring and a three-carbon chain with a ketone group. This complexity aids in creating regions of electron-rich and electron-deficient areas within the molecule, which is beneficial for the formation of charge-separated states and efficient photovoltaic processes. - **Integration into Larger Structure**: Motif 1 provides a suitable backbone around which additional functional groups and motifs can be attached, enhancing the overall photovoltaic performance by balancing electronic and structural properties.

Therefore, based on the importance of the thiophene group and the structural features that support photovoltaic efficiency, Motif 1 is chosen as the root motif.

---

Comments by Expert

---

correct, GPT answered motif 1, which obtains a distinct thiolphene unit that's constantly exist in conducting polymer and a lot of photovoltaic applications

---

Prompt

---

I want you to perform a simple post-processing step of the following response: "'Here's the detailed analysis of each motif in the given photovoltaic molecule, broken down:

1. **Motif 0**: - **Structure**: It is a 1,3-dithiol-2-one structural motif. It features a ketone (carbonyl group) with a five-membered ring containing two sulfur atoms adjacent to the ketone group.

2. **Motif 1**: - **Structure**: This motif consists of two distinct substructures: 1. A five-membered aromatic ring resembling thiophene, composed of a sulfur atom (yellow) and four carbon atoms. 2. A simple three-carbon chain with a double bond, ending in a ketone group (carbonyl group).

...'" The input is a response from another language agent. It describes motifs numbered from Motif 0 to Motif "'10'", inclusive! I want you to rephrase each motif description by filling in X within the following sentence template: This motif is X Be sure to condense the description and output a single PHRASE such that the sentence template is grammatically correct. Don't capitalize the first letter, since your answer should just be a phrase. Output your rephrasing for each motif on a SEPARATE line, using only a new line for delimiting different motifs. Don't output anything else. MAKE SURE you do it for EVERY MOTIF!

---

Response

---

1. a 1,3-dithiol-2-one structural motif with a ketone and a five-membered ring containing two sulfur atoms
2. two distinct substructures: a thiophene-like five-membered aromatic ring and a three-carbon chain ending in a ketone group
3. three different heterocyclic aromatic rings: thiophene, furan, and pyrrole
4. similar to Motif 0 but with an additional carbonyl group branching off the central carbon
5. a stylized thioacetone structure with a three-carbon chain and a sulfur atom connected to the central carbon
6. an indole-like structure with a benzene ring fused to a five-membered ring containing nitrogen and sulfur
7. a thioketo structure with a three-carbon chain and a sulfur atom double-bonded to the central carbon
8. a nitrile ketone structure with a three-carbon chain, a nitrile group, and a carbonyl group
9. a simple nitrile with a three-carbon chain and a nitrogen atom triple-bonded to the central carbon
10. an acetyl cyanide-like structure with a three-carbon chain, a ketone group, and a cyanide group
11. an isoprene structure with a four-carbon chain and two double bonds making up a conjugated diene

---

Comments by Expert

---

correct, gives precise description for each motif

---

Prompt

---

I want you to think like a chemist performing a detailed analysis of the chemical composition of a *photovoltaic molecule* through its constituent motifs. This requires two steps: 1) analyzing the individual motifs and 2) analyzing the pairwise interactions of motifs. The first step is already done. The second step is where I need your help. I will highlight for you different motif interactions within the same molecule. These interactions are numbered one-by-one, beginning with 0. Here are the textual descriptions of each motif interaction pair.

Interaction 0 features "'1. a 1,3-dithiol-2-one structural motif with a ketone and a five-membered ring containing two sulfur atoms'" and "'2. two distinct substructures: a thiophene-like five-membered aromatic ring and a three-carbon chain ending in a ketone group.'" Interaction 1 features "'2. two distinct substructures: a thiophene-like five-membered aromatic ring and a three-carbon chain ending in a ketone group'" and "'3. three different heterocyclic aromatic rings: thiophene, furan, and pyrrole.'" Interaction 2 features "'3. three different heterocyclic aromatic rings: thiophene, furan, and pyrrole and 1. a 1,3-dithiol-2-one structural motif with a ketone'" and a "'five-membered ring containing two sulfur atoms'".

I want you to tell me which one of these is MOST important, and which one of these is LEAST important. Output one number identifying the MOST important, and give a brief explanation. Output one number identifying the LEAST important, and a brief explanation why.

Response

Given the importance of the motifs and their interactions in photovoltaic molecules, let's consider the roles played by each interaction.

**Most Important Interaction (Number 1): 2** **Reasoning:** Interaction 2 features three heterocyclic aromatic rings: thiophene, furan, and pyrrole, interacting with a 1,3-dithiol-2-one structural motif with a ketone and a five-membered ring containing two sulfur atoms. This interaction involves a larger number of functional groups with a variety of heteroatoms, contributing significantly to the electronic properties and potential charge transfer efficiency. The presence of multiple aromatic rings could enhance electron delocalization, improving the photovoltaic properties of the molecule.

**Least Important Interaction (Number 0): 1** **Reasoning:** Interaction 1 involves a 1,3-dithiol-2-one structural motif interacting with two substructures: a thiophene-like five-membered aromatic ring and a three-carbon chain ending in a ketone group. While the 1,3-dithiol-2-one motif is crucial, the three-carbon chain ending in a ketone group is relatively straightforward and less complex compared to the multiple heterocyclic aromatic systems presented in interactions 1 and 2. The lack of additional aromatic heterocycles in this interaction makes it less critical in the context of electronic interactions within the molecule.

Thus, the most important interaction is **2**, and the least important interaction is **1**.

Comments by Expert

correct, interaction 2 is the most important " features three heterocyclic aromatic rings: thiophene, furan, and pyrrole, interacting with a 1,3-dithiol-2-one structural motif with a ketone and a five-membered ring containing two sulfur atoms" because interaction 1 carries the most essential photovoltaic functional groups in this molecule, and interaction 1 is the least important (the most lacking of additional aromatic cycles among the listed three interactions)

### A.2.1 EXAMPLE DESIGN STORY

We provide an example design story for CN1C(=O)c2c3c(c(C(F)(F)F)c(-c4c5C(=O)[Si](C)(C)C(=O)c5cn4C)c4C(=O)N(C)C(O)c(c43)c(C(F)(F)F)c2)C1=O. The story begins by rationalizing the choice of a root motif, in the context of designing organic photovoltaics,

then proceeds to explain each positive-negative connection in the parse tree from the perspective of structure-property relationships and design. Here is the story:

1. For the monomer described, intended for organic photovoltaic applications, the most essential motif would be the one that best supports charge transport and absorption of light. In organic photovoltaics, aromatic and conjugated systems are crucial because they enable extended $\pi$-conjugation, which improves the absorption of solar radiation and facilitates the transport of charge carriers.

Among the motifs provided, the **Pyrimidine ring (number 2)** is a strong candidate due to its aromatic nature and the presence of nitrogen atoms. The nitrogen atoms in a pyrimidine ring can participate in hydrogen bonding and other electroactive interactions, making this structure very important for the electronic properties of the polymer.

### Therefore, the root motif is **number 2**.

#### Reasoning:

- **Conjugation and Aromaticity**: The pyrimidine ring is aromatic, providing a structure that allows for delocalization of $\pi$-electrons, which is beneficial for light absorption and charge transport.
- **Nitrogen Atoms**: The nitrogen atoms can engage in various intermolecular interactions which are important for altering the electronic properties and improving the efficiency in photovoltaic applications. - **Versatility**: Pyrimidine rings are known to be versatile components in organic electronics, capable of facilitating charge carrier mobility and stability.

Thus, the Pyrimidine ring stands out as the single most important motif essential to the chemical profile of the monomer for its usefulness in organic photovoltaic applications. 2. The positive (red) and negative (green) motifs in this polymer design play a crucial role in enhancing its electronic properties, making it a viable candidate for applications such as organic solar cells. The red motif, likely a conjugated system with extended pi-electron delocalization, serves as the electron-donating or transporting component. The green motif, an electron-withdrawing group, is strategically positioned to create a strong dipole moment and facilitate charge separation. This donor-acceptor interplay is justified as it maximizes the polymer's ability to harness photogenerated excitons and promotes efficient charge transfer. The integration of the negative motif within the red motif structure lowers the overall bandgap, which is beneficial for the absorption of a broader spectrum of sunlight, thus increasing the efficiency of solar energy conversion. Furthermore, the inherent stability and solubility imparted by this design ensure that the polymer can be processed in various thin-film fabrication techniques, making it highly adaptable for use in organic photovoltaics. 3. In the design of polymer materials for applications such as organic solar cells, the positive (red) and negative (green) motifs play crucial roles in optimizing the photophysical and electronic properties of the polymer. The red motif, an aromatic structure with electron-donating groups, enhances the polymer's ability to absorb sunlight and facilitate exciton generation. The green motif, an adjacent electron-withdrawing unit, increases the polymer's electron affinity, aiding in the efficient separation and transport of charge carriers upon photoexcitation. This juxtaposition of donor and acceptor segments within the same molecular framework creates a built-in push-pull system, improving charge separation efficiency and minimizing recombination losses. Such finely tuned electronic interactions are essential for achieving high power conversion efficiencies in organic solar cells, as they directly impact the material's photovoltaic performance. Thus, the careful integration of these motifs is justified as it provides a strategic means to balance and enhance the photovoltaic and charge transport properties needed for high-performance organic solar materials. 4. The incorporation of the negative (green) motif into the positive (red) motif within this polymer design is pivotal for enhancing its functionality in specific applications like organic solar cells. The positive motif, marked in red, likely represents an electron-rich region, while the negative motif, highlighted in green, signifies an electron-deficient segment. This strategic juxtaposition creates a donor-acceptor (D-A) interaction, which is essential for effective charge separation and transport. This configuration helps facilitate the alternating push-pull effect of electrons, which is crucial in improving the photophysical properties of the polymer, such as light absorption and charge carrier mobility. In organic solar cells, such D-A conjugated systems enhance the absorption spectrum, increase the efficiency of exciton dissociation, and promote the transport of electrons and holes, thereby improving the overall power conversion efficiency. Therefore, the careful design and synthesis of these alternating motifs are justified as they provide a molecular architecture conducive to high-performance organic photovoltaic materials. 5. The positive (red) and negative (green) motifs in the polymer depicted play a crucial role

in determining its electronic properties and overall efficacy in applications like organic solar cells. The positive motif likely represents an electron-rich segment, while the negative motif represents an electron-deficient segment. This complementary interaction promotes intramolecular charge transfer, which is essential for enhancing the polymer's ability to absorb light and generate electron-hole pairs efficiently. The strategic incorporation of the green motif within the red framework ensures a balanced electronic structure, thereby optimizing the separation of charge carriers and minimizing recombination losses. This facilitates effective charge transport, leading to improved photovoltaic performance. Furthermore, this donor-acceptor architecture can be tuned to adjust the absorption spectrum, enabling better alignment with the solar spectrum. Hence, the addition of the negative (green) motif to the positive (red) motif is a deliberate design choice to enhance the polymer's semiconducting properties, making it highly effective for applications in organic solar cells. 6. In the design of polymers for organic solar cells, the strategic incorporation of the negative (green) motif into the positive (red) motif is crucial for optimizing the electronic properties and enhancing performance. The positive motif, depicted in red, likely represents an electron-donating unit, while the negative motif, shown in green, is indicative of an electron-withdrawing unit. When these two motifs are combined within the polymer backbone, they create an alternating donor-acceptor structure that facilitates charge separation and transport by lowering the bandgap and increasing the polymer's ability to absorb sunlight. This push-pull interaction between the electron-rich and electron-deficient segments contributes to the creation of effective pathways for exciton dissociation and charge carrier mobility. Consequently, this interaction plays a fundamental role in balancing and optimizing the absorption spectrum, charge transport, and overall efficiency of the organic solar cells, making the addition of the green motif justified and beneficial for enhancing the performance of such devices. 7. In the design of polymers for organic solar cells, the interaction between the positive (red) and negative (green) motifs is of paramount importance for optimizing the electronic properties of the material. The positive motif, marked in red, likely represents an electron-donating unit, while the negative motif, marked in green, represents an electron-accepting unit. This push-pull mechanism, also referred to as a donor-acceptor interaction, is crucial for improving charge separation and transport within the polymer. By incorporating both electron-donating and electron-withdrawing motifs, the resulting polymer exhibits a narrowed bandgap, enhancing its ability to absorb a broader spectrum of sunlight. The dipole moments generated at the interface between these motifs also facilitate exciton dissociation, improving the efficiency of charge generation. Additionally, the conjugated backbone formed by the alternating positive and negative motifs creates a pathway for electron mobility, which is essential for the polymer's performance as an active layer in organic solar cells. Thus, the deliberate design incorporating these specific motifs is justified by its direct contribution to improved light absorption, charge separation, and overall photovoltaic efficiency. 8. The design of the polymer shown, which includes alternating positive (red) and negative (green) motifs, is likely aimed at enhancing the material's electronic properties, useful for applications such as organic solar cells. The positive motif suggests regions with electron-withdrawing characteristics while the negative motif indicates electron-donating properties. This juxtaposition creates a donor-acceptor interaction framework that is crucial for tuning the polymer's bandgap. In organic solar cells, such a tailored bandgap is vital for effective light absorption and charge separation. The electron-rich (negative) motif can facilitate enhanced electron mobility while the electron-deficient (positive) motif can improve hole transport efficiency. This duality ensures the polymer can effectively separate and transport charge carriers generated upon light absorption, thereby improving the overall efficiency of the solar cells. The inclusion of side chains and functional groups further modulates solubility, processability, and morphological stability, making the polymer system adaptable for practical device fabrication. 9. In the context of designing polymers for applications such as organic solar cells, the integration of a negative (green) motif with a positive (red) motif is crucial for optimizing the electronic properties and ensuring effective performance. The red motif in the polymer structure typically represents electron-donating units, which enhance the electron density and facilitate charge transport. The green motif, on the other hand, represents electron-withdrawing units, which help to lower the polymer's overall energy levels and improve electron acceptance. This positive-negative interaction is fundamental because it results in a defined energy band gap that is essential for efficient absorption of sunlight and conversion into electrical energy. Additionally, the interplay between these motifs enhances the polymer's ability to form a well-ordered, crystalline structure, further improving charge mobility and overall efficiency. Their balanced interaction improves the stability, solubility, and mechanical properties of the polymer, making it suitable for real-world applications in organic electronics and photovoltaics. 10. In designing a polymer for applications like organic solar cells, the interaction between the positive (red) and negative (green) motifs plays a

critical role in modulating the polymer's electronic properties. The positive red motif, in this case, forms the backbone of the polymer's structure, contributing to the overall electronic conduction. By introducing the negative green motif, which likely contains electron-withdrawing groups such as fluorine atoms, the electron density distribution within the polymer is altered, enhancing its ability to facilitate charge transfer processes. This complementary interaction can enhance the polymer's capacity for efficient charge separation and transport, crucial for the effective conversion of light to electricity in organic solar cells. Furthermore, the specific electron-donating and electron-accepting properties of these motifs can be fine-tuned to optimize the polymer's absorption spectrum and energy levels, making the material more effective in capturing sunlight and transforming it into usable energy. Thus, the integration of these motifs is justified through a detailed understanding of their synergistic effects on the polymer's photovoltaic performance. 11. The polymer depicted showcases a strategic combination of a positive red motif and a negative green motif, designed to enhance its properties for applications like organic solar cells. The red motif, rich in electronegative carbonyl (C=O) and fluorine (F) groups, likely contributes to the electron-accepting characteristics necessary for efficient charge separation and transfer, critical in photovoltaic functions. Meanwhile, the green motif, embedded centrally, presumably acts as an electron-donating unit due to its conjugated ring structure with additional electron-withdrawing carbonyl groups enhancing the polymer's ability to create a stable, low-energy LUMO level. The interplay between these motifs through $\pi$-conjugation and potential intramolecular charge transfer enhances the polymer's electronic properties, such as its photochemical stability, light absorption, and charge mobility. This synergistic interaction justifies the inclusion of the negative motif to optimize the polymer's efficiency in harvesting solar energy, delivering performance enhancements critical for next-generation organic solar cells. 12. In evaluating the design of the polymer for potential applications such as organic solar cells, the interaction between the positive (red) and negative (green) motifs is crucial. The strategic incorporation of the negative motif, marked by its electron-withdrawing groups (such as the carbonyl groups), is critical in creating an internal charge transfer within the polymer matrix. This enhances electron mobility by facilitating a donor-acceptor interaction where the red motif acts as the electron donor and the green motif functions as the electron acceptor. Such a configuration supports efficient separation and transport of charge carriers. Consequently, this charge transfer interaction reduces recombination losses and increases the overall efficiency of the polymer in solar energy conversion. Additionally, the spatial orientation and electronic properties of the combined motifs influence the polymer's absorption spectrum and photophysical properties, optimizing light absorption. Thus, the deliberate addition of the negative motif to the polymer structure is justified by its substantial impact on enhancing electrical conductivity, charge separation, and optical properties, which are vital for the efficacy and performance of organic solar cells. 13. In the context of designing a polymer for applications such as organic solar cells, the incorporation of positive and negative motifs plays a pivotal role in tailoring the material's electronic properties. The positive (red) motif may be indicative of an electron-donating moiety, which can enhance the charge carrier mobility within the polymer. Conversely, the negative (green) motif likely represents an electron-withdrawing group, crucial for stabilizing the electron density and improving the polymer's electron affinity. The interaction between these motifs creates a push-pull effect in the polymer backbone, which is vital for optimizing the absorption of a broad spectrum of light and facilitating efficient charge separation and transport. This push-pull mechanism is particularly advantageous in organic solar cells, as it contributes to higher power conversion efficiency by maximizing the generation and transport of free charge carriers when the polymer is exposed to sunlight. The strategic placement and balance of these motifs enable fine-tuning of the polymer's HOMO-LUMO (highest occupied molecular orbital-lowest unoccupied molecular orbital) gap, ensuring it is suitable for effective photovoltaic performance. 14. In the given polymer, the design incorporates a positive (red) and a negative (green) motif, which plays a critical role in its chemical and physical properties. The positive motif, marked in red, is likely an electron-donor segment that provides a source of electrons through conjugated systems or electron-rich groups. The negative motif, highlighted in green, could incorporate electron-withdrawing groups that facilitate electron acceptance, making it an electron-acceptor segment. This complementary interaction between donor and acceptor segments within the polymer is fundamental for optimizing the charge-transfer processes, which are essential in applications such as organic solar cells. When light excites the polymer, the positive (red) motif can donate electrons that are efficiently transferred to the negative (green) motif. This charge separation is crucial for generating electrical current in organic photovoltaics. Additionally, this positive-negative interaction can enhance the polymer's stability, morphology, and overall electronic properties, making it a viable candidate for high-performance organic solar cells. 15. In designing polymers for applications like

organic solar cells, the interaction between the positive (red) and negative (green) motifs plays a critical role in optimizing the polymer's electronic properties and overall performance. The red motif, likely possessing electron-donating characteristics, enhances the polymer's ability to transport holes, making it an efficient donor material. On the other hand, the green motif, characterized by electron-withdrawing properties, contributes to electron transport and serves as an acceptor material. The juxtaposition of these contrasting electronic features within the same polymer backbone facilitates effective charge separation and transport, crucial for the efficiency of organic photovoltaic devices. Moreover, the fine-tuning of these donor-acceptor interactions influences the polymer's bandgap and energy levels, which can be tailored to maximize light absorption in the solar spectrum, thus enhancing the photocurrent generation in solar cell applications. This delicate balance and interaction between the positive and negative motifs thereby justify their integration, significantly contributing to the polymer's optoelectronic properties and making it a viable candidate for high-performance organic solar cells. 16. In designing polymers for applications such as organic solar cells, the interplay between electron-donating (red) and electron-withdrawing (green) motifs is pivotal for optimizing the electronic properties and stability of the material. The red motif serves as an electron-donating unit, facilitating the creation of a high-energy orbital system essential for effective light absorption and exciton generation. Conversely, the green motif acts as an electron-accepting unit, which helps in stabilizing the generated excitons and improving charge separation efficiency. This complementary interaction between the electron-rich and electron-deficient segments creates a balanced distribution of electronic density, thereby fine-tuning the energy levels and improving charge transport abilities. Additionally, this donor-acceptor synergy enhances the structural rigidity and thermal stability of the polymer, making it more robust under operational conditions. The judicious incorporation of the negative (green) motif to the positive (red) motif is thus essential in engineering polymers with the desirable electronic and physical characteristics suited for high-performance organic solar cells. 17. In the context of designing a polymer for applications such as organic solar cells, the interaction between the positive (red) and negative (green) motifs is crucial for tuning the polymer's optoelectronic properties. The negative motif in green, characterized by electron-withdrawing groups, enhances the polymer's electron affinity, improving its ability to accept electrons. This leads to a lower energy band gap, which is favorable for absorbing a broader spectrum of sunlight. Additionally, the complementary positioning of these motifs enhances charge separation and transport efficiencies within the polymer matrix. The electron-rich positive motif in red can act as a donor, facilitating effective charge transfer processes when paired with the electron-deficient negative motif. By strategically incorporating these motifs into the polymer framework, we can optimize the material's photovoltaic performance, achieving better light absorption, higher charge carrier mobility, and ultimately improved efficiency in converting solar energy to electrical energy in organic solar cells. 18. In the context of designing polymers for applications such as organic solar cells, the positive (red) and negative (green) motifs are strategically integrated to optimize electronic and structural properties. The red motif, being a conjugated aromatic structure, offers high electron density and good charge transport characteristics due to its delocalized $\pi$-electrons, which is crucial in facilitating efficient charge mobility. Conversely, the green motif, characterized by its electron-withdrawing functional groups (e.g., carbonyl and fluorine atoms), introduces electron deficiency into the polymer chain. This electron-withdrawing nature helps to lower the polymer's LUMO (Lowest Unoccupied Molecular Orbital) energy level, aiding in the enhancement of electron acceptor properties. The presence of both motifs creates a donor-acceptor (D-A) interaction within the polymer, optimizing the solar cell's intrinsic properties such as bandgap tuning, absorption spectrum, and charge separation efficiency. This deliberate juxtaposition of electron-rich and electron-deficient motifs forms a polymer network that is highly suitable for converting sunlight into electrical energy with maximized efficiency, making this motif combination crucial for advanced organic photovoltaic applications. 19. In the design of a polymer for applications such as organic solar cells, the incorporation of complementary electronic motifs is crucial for optimizing charge transfer and enhancing device efficiency. The positive motif highlighted in red and the negative motif in green represent electron-donating and electron-withdrawing segments, respectively. This donor-acceptor architecture facilitates effective intermolecular charge transfer dynamics, which is essential for efficient exciton dissociation and charge transport within the polymer matrix. The electron-donating capacity of the red motif, typically featuring conjugated systems and groups that can delocalize electrons, complements the electron-deficiency of the green motif, often imbued with electronegative groups or atoms like fluorine and carbonyl groups. This interaction not only promotes optimal energy level alignment between the highest occupied molecular orbital (HOMO) and the lowest unoccupied molecular orbital (LUMO) but also enhances the polymer's photophys-

ical properties by broadening its absorption spectrum. Such a synergistic design is beneficial for increasing the power conversion efficiency of organic solar cells by maximizing light absorption and facilitating efficient charge separation and mobility. 20. In the context of designing polymers for applications such as organic solar cells, the interplay between electron-rich (positive, red) and electron-deficient (negative, green) motifs is of paramount importance. The red motif, characterized by its extended $\pi$-conjugation, acts as an electron-donating unit, facilitating efficient charge transport. Meanwhile, the green motif introduces electron-withdrawing functionalities, thereby reducing the polymer's highest occupied molecular orbital (HOMO) energy levels while increasing its lowest unoccupied molecular orbital (LUMO) energy levels. This complementary pairing creates a unique donor-acceptor interface within the polymer structure, enhancing charge separation and thereby improving the photovoltaic performance. Furthermore, the electron-withdrawing groups can stabilize the resulting negative charges, reducing charge recombination rates. This careful juxtaposition of motifs thereby optimizes light absorption, enhances charge carrier mobility, and ultimately leads to improved energy conversion efficiencies in organic solar cell applications. 21. In designing polymers for applications such as organic solar cells, the strategic incorporation of both positive (red) and negative (green) motifs is essential to optimize the material's properties. The red motif, a polyaromatic segment, serves as an electron donor, while the green motif, with silicon and carbonyl groups, acts as an electron acceptor due to its electron-withdrawing nature. This donor-acceptor interaction enhances charge separation and charge carrier mobility within the polymer, crucial for efficient photovoltaic performance. The juxtaposition of these motifs can lead to a reduction in the polymer's band gap, increased light absorption, and improved exciton dissociation, thereby enhancing the efficiency of organic solar cells. Additionally, the specific arrangement of the motifs affects the crystallinity and morphological stability of the polymer, which are pivotal for device performance and longevity. This synergistic design illustrates how the confluence of electron-rich and electron-deficient units can be exploited to tailor the electronic and physical properties of polymers, rendering them suitable for cutting-edge applications in organic electronics.

## A.3 CASE STUDY: ISOCYANATES

Isocyanates are highly reactive organic compounds characterized by the presence of one or more isocyanate groups (-N=C=O). Their high reactivity makes them essential in the production of polyurethanes, this reaction forms urethane linkages, which are the backbone of polyurethane materials. Polyurethanes that are synthesized from isocyanates are really versatile, with a wide range of applications from flexible foams and elastomers to rigid insulation materials. The molecular structure of the isocyanate can be tailored, with aliphatic or aromatic variations influencing the final properties of the polymer, such as its mechanical strength, flexibility, and resistance to environmental factors like UV radiation. However, due to their high reactivity, isocyanates are associated with potential health hazards, including irritation and sensitization. The safety concerns attached to this compounds made it requires strict industrial safety protocols during handling and processing.

For isocyanate, features -N=C=O and highly reactive, especially in forming polyurethane linkages. The selected molecule contains two distinct -N=C=O groups serving as core functional groups to form isocyanate and highly represent the desired reactivity in this dataset of the molecules.

| Prompt |
|--------|

1836
1837
1838
1839
1840
1841
1842
1843
1844
1845
1846
1847
1848
1849
1850
1851
1852
1853
1854
1855
1856
1857
1858
1859
1860
1861
1862
1863
1864
1865
1866
1867
1868
1869
1870
1871
1872
1873
1874
1875
1876
1877
1878
1879
1880
1881
1882
1883
1884
1885
1886
1887
1888
1889

I want you to think like a chemist performing a detailed analysis of the chemical composition of an isocyanates through its constituent motifs. I will highlight for you 9 of the substructures of a molecule. They are numbered one-by-one from Motif 0 to Motif 8, inclusive. I want you to explain, concisely, what each numbered motif is. Make sure to start from Motif 0 and go in order of the numbering. MAKE SURE you describe EVERY MOTIF!

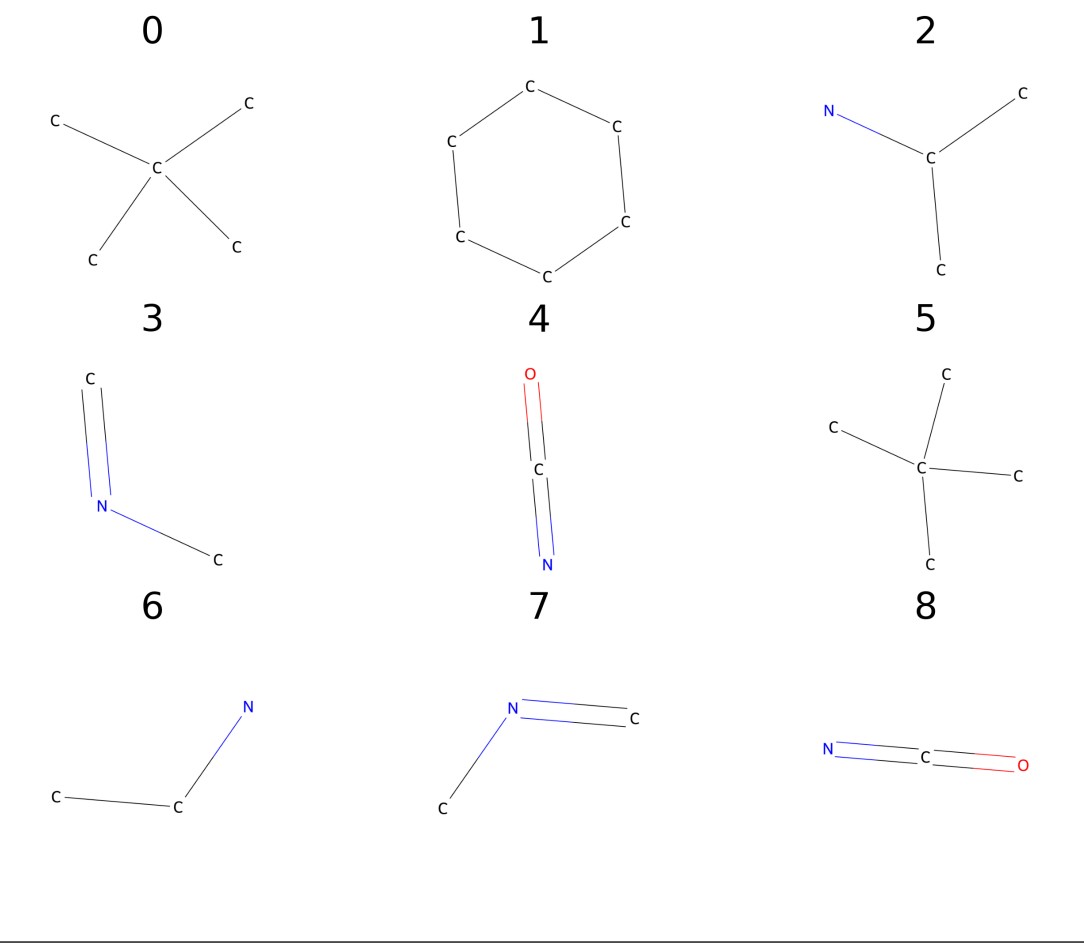

Response

Sure, let's analyze each motif in order:

%%% Motif 0 This structure represents a tert-butyl group. It consists of a central carbon atom bonded to three methyl groups (each a CH3 group).

%%% Motif 1 This structure is a cyclohexane ring. It consists of six carbon atoms arranged in a hexagonal ring, with each carbon atom bonded to two other carbons and two hydrogen atoms (not shown).

%%% Motif 2 This structure is an ethylamine group. It consists of an amine group (NH2) attached to an ethyl group (CH3-CH2-).

%%% Motif 3 This structure is an ethylideneamine motif. It consists of a carbon-nitrogen (C=N) double bond, with the nitrogen also bonded to an alkyl group (CH3) and the carbon to another carbon.

%%% Motif 4 This structure features an O=C=N group, indicating the presence of an isocyanate group bonded to a carbonyl group. This is a key motif in many urea and carbamate derivatives.

%%% Motif 5 This structure represents a neopentyl group. It consists of a central carbon atom bonded to four other carbon atoms: one in a methyl group (CH3) and three in methylene groups (CH2).

%%% Motif 6 This structure shows a nitrile group (C≡N) attached to an ethyl group (CH3-CH2-). Nitriles are commonly found in organic compounds.

%%% Motif 7 This structure is a cyanoacetamide group. It consists of a nitrile group (C≡N) adjacent to a carbonyl group (C=O).

%%% Motif 8 This structure is an isocyanate group (N=C=O). It consists of a nitrogen atom double-bonded to a carbon atom, which is also double-bonded to an oxygen atom.

These motifs may represent fragments of larger molecules, often seen in various industrial and chemical synthesis processes.

---

Comments by Expert

---

Easy prompt, clear restrictions and instructions GPT's answer: all correct, clearly stated the functionality of each motif from 0 to 8.

---

Prompt

I want you to think like a chemist performing a detailed analysis of the chemical composition of an isocyanates through its constituent motifs. I will highlight for you some of the distinctive fragments of a molecule. They are numbered from 0 and individually highlighted in RED. Focus ONLY on the substructure highlighted in red within each cell. Here is the descriptions for each substructure provided by an expert:

Sure, let's analyze each motif in order:

%%% Motif 0 This structure represents a tert-butyl group. It consists of a central carbon atom bonded to three methyl groups (each a CH3 group).

%%% Motif 1 This structure is a cyclohexane ring. It consists of six carbon atoms arranged in a hexagonal ring, with each carbon atom bonded to two other carbons and two hydrogen atoms (not shown).

%%% Motif 2 This structure is an ethylamine group. It consists of an amine group (NH2) attached to an ethyl group (CH3-CH2-).

%%% Motif 3 This structure is an ethylideneamine motif. It consists of a carbon-nitrogen (C=N) double bond, with the nitrogen also bonded to an alkyl group (CH3) and the carbon to another carbon.

%%% Motif 4 This structure features an O=C=N group, indicating the presence of an isocyanate group bonded to a carbonyl group. This is a key motif in many urea and carbamate derivatives.

%%% Motif 5 This structure represents a neopentyl group. It consists of a central carbon atom bonded to four other carbon atoms: one in a methyl group (CH3) and three in methylene groups (CH2).

%%% Motif 6 This structure shows a nitrile group (C≡N) attached to an ethyl group (CH3-CH2-). Nitriles are commonly found in organic compounds.

%%% Motif 7 This structure is a cyanoacetamide group. It consists of a nitrile group (C≡N) adjacent to a carbonyl group (C=O).

%%% Motif 8 This structure is an isocyanate group (N=C=O). It consists of a nitrogen atom double-bonded to a carbon atom, which is also double-bonded to an oxygen atom.

These motifs may represent fragments of larger molecules, often seen in various industrial and chemical synthesis processes.

I want you to tell me if any two of them should be combined together to form a more meaningful substructure. Remember the defining isocyanate group is -N=C=O. Your task is to highlight the primary functional groups of the molecule. Output a single pair of numbers if you think those two fragments should be combined, and a brief explanation why. If no such pairs exist, don't output anything.

---

Response

---

%%% Motif 4 This structure features an O=C=N group, representing the presence of an isocyanate group bonded to a carbonyl group, which is a critical motif in many urea and carbamate derivatives.

%%% Motif 8 This structure features an isocyanate group (N=C=O), consisting of a nitrogen atom double-bonded to a carbon atom, which is also double-bonded to an oxygen atom.

Combining **Motif 4** and **Motif 8** forms the primary functional group: a carbonyl group directly attached to an isocyanate group, which is critical for identifying the chemical properties and reactivity of isocyanates.

%%% **Conclusion:** **4, 8** may be combined into a functional group consisting of a carbonyl bonded directly to an isocyanate, making the structure more meaningful in the context of isocyanate chemistry.

---

Comments by Expert

---

Easy, asking GPT what two groups should be combined to form isocyanate GPT's answer is 4 and 8. GPT is correct to point out both are isocyanates already, but not correct to say they should be combined

Prompt

I want you to think like a chemist performing a detailed analysis of the chemical composition of an isocyanates through its constituent motifs. I will highlight for you some of the distinctive substructures of an isocyanates. They are numbered from 0. Here are the textual descriptions of each motif:

Sure, let's analyze each motif in order:

%%% Motif 0 This structure represents a tert-butyl group. It consists of a central carbon atom bonded to three methyl groups (each a CH3 group).

%%% Motif 1 This structure is a cyclohexane ring. It consists of six carbon atoms arranged in a hexagonal ring, with each carbon atom bonded to two other carbons and two hydrogen atoms (not shown).

%%% Motif 2 This structure is an ethylamine group. It consists of an amine group (NH2) attached to an ethyl group (CH3-CH2-).

%%% Motif 3 This structure is an ethylideneamine motif. It consists of a carbon-nitrogen (C=N) double bond, with the nitrogen also bonded to an alkyl group (CH3) and the carbon to another carbon.

%%% Motif 4 This structure features an O=C=N group, indicating the presence of an isocyanate group bonded to a carbonyl group. This is a key motif in many urea and carbamate derivatives.

%%% Motif 5 This structure represents a neopentyl group. It consists of a central carbon atom bonded to four other carbon atoms: one in a methyl group (CH3) and three in methylene groups (CH2).

%%% Motif 6 This structure shows a nitrile group (C≡N) attached to an ethyl group (CH3-CH2-). Nitriles are commonly found in organic compounds.

%%% Motif 7 This structure is a cyanoacetamide group. It consists of a nitrile group (C≡N) adjacent to a carbonyl group (C=O).

%%% Motif 8 This structure is an isocyanate group (N=C=O). It consists of a nitrogen atom double-bonded to a carbon atom, which is also double-bonded to an oxygen atom.

These motifs may represent fragments of larger molecules, often seen in various industrial and chemical synthesis processes.

I want you to pick only ONE of these as the root motif most essential to its chemical profile. It should be the single most important motif the rest of the isocyanate was built around. The defining isocyanate group is -N=C=O, so your selected root motif MUST contain the group. If there are multiple such motifs, or one doesn't clearly stand out, just pick one of them. Give your answer as a single number. Explain your reasoning carefully.

---

Response

---

%%% Answer: 8

%%% Reasoning:

In the given motifs, we are looking for the motif that contains the isocyanate group (-N=C=O) at its core, as this functional group is essential to the chemical profile of isocyanates.

Among the given motifs, Motif 8 includes the isocyanate group distinctly:

- Motif 8: It features the isocyanate group (N=C=O) prominently and without any additional structural complexity directly attached to it. This makes it the simplest and most straightforward representation of the essential isocyanate entity.

This feature, the critical and defining part of an isocyanate, means Motif 8 is the most crucial to the chemical profile of the substance. The rest of the molecule is built around this group, making it the root structure.

Other motifs either have additional complex groups attached to the isocyanate group or other functional groups that could potentially classify the molecules differently. Hence, Motif 8 is chosen as it aligns precisely with the core chemical characteristic we are focusing on.

---

Comments by Expert

---

Medium prompt, since picking the most important functional group involves empirical experience in organic chemistry, and the answer required one individual selection can bring discrepancies sometimes between language models and chemistry expert. GPT's answer: correct, GPT answer motif 8 since it features the distinct isocyanate group. GPT even clearly pointed out that other motifs either have additional complex groups attached to the isocyanate group or other functional groups that could potentially classify the molecules differently. It gives perfect answer to this question.

## A.4 CASE STUDY: ACRYLATES

Acrylates refer to a broad class of chemical compounds that are derivatives of acrylic acid and its related esters. They can undergo polymerization rapidly through a free-radical mechanism, making them particularly useful in applications requiring fast curing processes, such as in adhesives, coatings, sealants, and 3D printing materials. Acrylates possess a double bond, which is highly reactive in the presence of free radicals, allowing them to form long polymer chains with varying degrees of crosslinking. This flexibility in polymer structure endow acrylates with unique properties such as optical clarity, UV resistance, and flexibility, depending on the specific formulation. In industries like automotive, aerospace, and biomedical devices, acrylate-based materials are featured for their robust performance in tough environments. Additionally, modified acrylates such as methacrylates are also featured for multifunctions and with greater control over polymers' properties including rigidity, toughness, and adhesion.

For acrylate, the molecule I selected have typical ester/acrylate functional groups, the rest of the part containing both carbon chains and aromatic structures. The molecule is not over complicated acrylate example but have all functional groups for GPT to learn and it's very representative. This molecule obtains vinyl end groups which are highly reactive in polymerization reactions, making it a good candidate for studying acrylate reactivity as well.

Prompt

I want you to think like a chemist performing a detailed analysis of the chemical composition of an acrylates through its constituent motifs. I will highlight for you 7 of the substructures of a molecule. They are numbered one-by-one from Motif 0 to Motif 6, inclusive. I want you to explain, concisely, what each numbered motif is. Make sure to start from Motif 0 and go in order of the numbering. MAKE SURE you describe EVERY MOTIF!

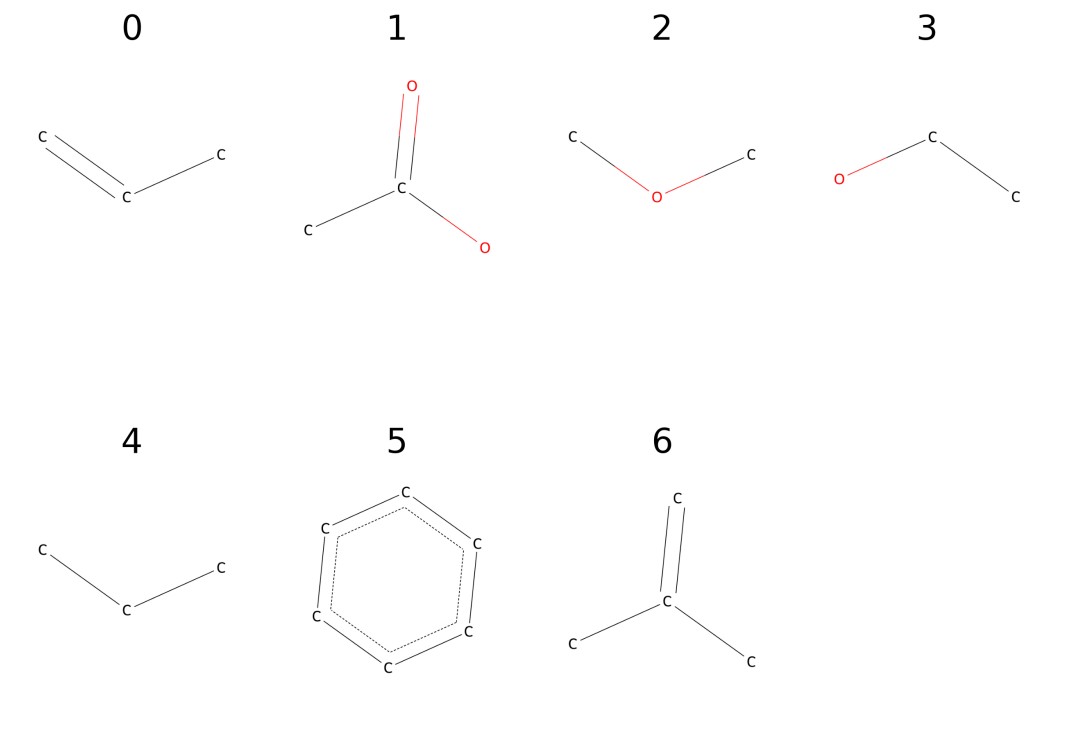

Response

Sure, here's a concise description of each numbered motif as per the image provided:

**Motif 0**: This motif is a vinyl group, which consists of two carbon atoms connected by a double bond (C=C), with each carbon atom having one additional connection for a total of three.

**Motif 1**: This is an acrylate group, also known as a propenoate ester. It consists of a carbonyl group (C=O) attached to a carbon atom, which also has an oxygen atom bonded to another carbon atom (an ester linkage).

**Motif 2**: This motif is an ethyl ether group, specifically an ether where an oxygen atom connects two ethyl (CH3-CH2) groups.

**Motif 3**: This is a methoxy group, where an oxygen atom is attached to a single methyl group (CH3).

**Motif 4**: This is a simple propyl group, consisting of three carbon atoms in a chain (CH3-CH2-CH2).

**Motif 5**: This is a benzene ring, also known as an aromatic ring, which is made up of six carbon atoms arranged in a hexagonal ring with alternating double bonds between carbon atoms.

**Motif 6**: This is an isopropyl group, a branched chain comprising three carbon atoms where the central carbon is connected to two methyl groups (CH3-CH-CH3).

Each motif represents a fundamental building block that can contribute to the structure and properties of an acrylate molecule.

Comments by Expert

Easy, really clear and straightforward prompt, clear restrictions to prohibit GPT making mistakes. Mentioning the total motif and following the order of motifs provide GPT clear logic chin to tackle this question. GPT's answer Correct for all motifs, clear explanation of each motif's functionality Prompt

I want you to think like a chemist performing a detailed analysis of the chemical composition of an acrylate through its constituent motifs. I will highlight for you some of the distinctive fragments of a molecule. They are numbered from 0 and individually highlighted in GREEN. Focus ONLY on the substructure highlighted in green within each cell. Here is the descriptions for each substructure provided by an expert:

Sure, here's a concise description of each numbered motif as per the image provided:

**Motif 0**: This motif is a vinyl group, which consists of two carbon atoms connected by a double bond (C=C), with each carbon atom having one additional connection for a total of three.

**Motif 1**: This is an acrylate group, also known as a propenoate ester. It consists of a carbonyl group (C=O) attached to a carbon atom, which also has an oxygen atom bonded to another carbon atom (an ester linkage).

**Motif 2**: This motif is an ethyl ether group, specifically an ether where an oxygen atom connects two ethyl (CH3-CH2) groups.

**Motif 3**: This is a methoxy group, where an oxygen atom is attached to a single methyl group (CH3).

**Motif 4**: This is a simple propyl group, consisting of three carbon atoms in a chain (CH3-CH2-CH2).

**Motif 5**: This is a benzene ring, also known as an aromatic ring, which is made up of six carbon atoms arranged in a hexagonal ring with alternating double bonds between carbon atoms.

**Motif 6**: This is an isopropyl group, a branched chain comprising three carbon atoms where the central carbon is connected to two methyl groups (CH3-CH-CH3).

Each motif represents a fundamental building block that can contribute to the structure and properties of an acrylate molecule.

I want you to tell me if any two of them should be combined together to form a more meaningful substructure. Remember the defining acrylate group is C=CC(=O)O, which consists of a carbon-carbon double bond and a carboxylate ester. Your task is to construct this defining acrylate group by combining the alkene or vinyl group with the ester group. If the acrylate group is already present, your task is to highlight any other noteworthy functional groups. Output a single pair of numbers if you think those two fragments should be combined, and a brief explanation why. If no such pairs exist, don't output anything.

Response

1 0. The reason for this particular combination is to form the defining acrylate group, C=CC(=O)O. By combining Motif 1 (the acrylate/propenoate ester featuring a carbonyl carbon bonded to an oxygen atom linked by an ester bond to another carbon) with Motif 0 (the vinyl group, an alkene with two carbon atoms connected by a double bond), we achieve the desired acrylate structure containing a carbon-carbon double bond and a carboxylate ester.

Comments by Expert

Correct, 0 and 1 should be combined together

Prompt

I want you to think like a chemist performing a detailed analysis of the chemical composition of an acrylates through its constituent motifs. I will highlight for you 6 of the substructures of a molecule. They are numbered one-by-one from Motif 0 to Motif 5, inclusive. I want you to explain, concisely, what each numbered motif is. Make sure to start from Motif 0 and go in order of the numbering. MAKE SURE you describe EVERY MOTIF!

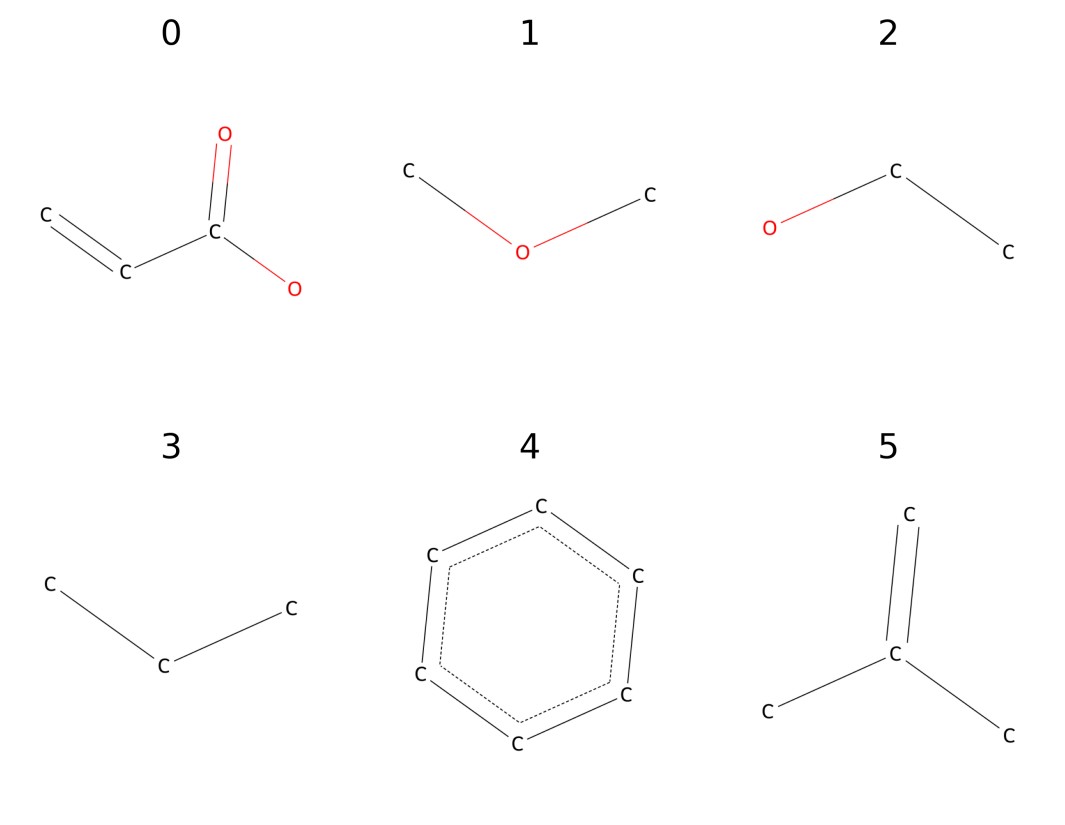

Response

Certainly, here is a concise characterization of each motif:

**Motif 0:** This is an ester group with an adjacent carbon-carbon double bond (C=C). The structure can be described as a vinyl ester derived from acrylic acid. Specifically, it contains a carbonyl group (C=O) adjacent to an oxygen single-bonded (O—C) to a vinyl group (—CH=CH2).

**Motif 1:** This is an ether group, characterized by an oxygen atom linked to two alkyl groups, which are represented here as carbon atoms (C—O—C). This motif shows the simplest ether functional group.

**Motif 2:** This is another ether group, but oriented differently in comparison to Motif 1. Here, an oxygen atom is bonded to two carbons in a linear fashion again forming the general formula of an ether (R—O—R').

**Motif 3:** This is an alkane chain with three carbons, indicating a simple linear hydrocarbon with single bonds between the carbon atoms (C—C—C). This motif is a fragment of a longer alkane chain or a part of more complex hydrocarbons.

**Motif 4:** This motif is a benzene ring, signified by its hexagonal structure with alternating double bonds. Benzene is an aromatic hydrocarbon with significant stability due to its conjugated pi electron system.

**Motif 5:** This is an isopropenyl group, described as an alkene in which the central carbon is double-bonded to one carbon and single-bonded to two other carbons (other than hydrogen). This motif presents a branching point in the structure with a double bond.

These motifs collectively depict a variety of functional groups and structural elements commonly found in the chemistry of acrylates and related organic compounds.

---

Comments by Expert

---

Correct

Prompt

---

ded by an expert:

Certainly, here is a concise characterization of each motif:

**Motif 0:** This is an ester group with an adjacent carbon-carbon double bond (C=C). The structure can be described as a vinyl ester derived from acrylic acid. Specifically, it contains a carbonyl group (C=O) adjacent to an oxygen single-bonded (O—C) to a vinyl group (—CH=CH2).

**Motif 1:** This is an ether group, characterized by an oxygen atom linked to two alkyl groups, which are represented here as carbon atoms (C—O—C). This motif shows the simplest ether functional group.

**Motif 2:** This is another ether group, but oriented differently in comparison to Motif 1. Here, an oxygen atom is bonded to two carbons in a linear fashion again forming the general formula of an ether (R—O—R').

**Motif 3:** This is an alkane chain with three carbons, indicating a simple linear hydrocarbon with single bonds between the carbon atoms (C—C—C). This motif is a fragment of a longer alkane chain or a part of more complex hydrocarbons.

**Motif 4:** This motif is a benzene ring, signified by its hexagonal structure with alternating double bonds. Benzene is an aromatic hydrocarbon with significant stability due to its conjugated pi electron system.

**Motif 5:** This is an isopropenyl group, described as an alkene in which the central carbon is double-bonded to one carbon and single-bonded to two other carbons (other than hydrogen). This motif presents a branching point in the structure with a double bond.

These motifs collectively depict a variety of functional groups and structural elements commonly found in the chemistry of acrylates and related organic compounds.

I want you to tell me if any two of them should be combined together to form a more meaningful substructure. Remember the defining acrylate group is C=CC(=O)O, which consists of a carbon-carbon double bond and a carboxylate ester. Your task is to construct this defining acrylate group by combining the alkene or vinyl group with the ester group. If the acrylate group is already present, your task is to highlight any other noteworthy functional groups. Output a single pair of numbers if you think those two fragments should be combined, and a brief explanation why. If no such pairs exist, don't output anything.

---

Response

---

For acrylate chemistry, the defining motif is C=CC(=O)O, which consists of a carbon-carbon double bond and a carboxylate ester group. Let's analyze the motifs:
**Motif 0:** - Contains the carbonyl group (C=O) and ester (O-R) parts, along with a carbon-carbon double bond, indicating it already represents an acrylate substructure.
**Motifs 1, 2, 3, 4, and 5:** - None of these motifs independently contain the defining features of an acrylate group. Motifs 1, 2, and 5 are ether and isopropenyl groups, while 3 is an alkane chain and 4 is a benzene ring.
**Result:** The acrylate group is already present in Motif 0, already encompassing both the ester and the vinyl group. There is no need to combine any other motifs to form the acrylate group, as it is already fully represented.
No pairs to combine, as the defining acrylate motif is already contained within Motif 0.

---

Comments by Expert

---

Fully correct, the defining motif is already contained

---

## A.5 Case Study: Chain Extenders

Chain extenders are small, difunctional molecules used in polymer chemistry to extend and link polymer chains, thereby increasing the molecular weight and enhancing the physical properties of the resulting polymer. The bonding process triggered by chain extenders lead to materials with improved mechanical properties, such as higher tensile strength, flexibility, and impact resistance. Chain extenders are particularly important in applications requiring tough yet flexible materials, such as elastomers, foams, and coatings. In polyurethane systems, for example, chain extenders typically have two reactive groups, such as hydroxyl or amine groups, which react with isocyanates or other functional groups to create strong, covalent bonds between polymer segments. The choice of chain extenders can hugely impact the final material properties; for example, the use of diamines versus diols in polyurethanes can significantly influence polymers' elasticity and thermal stability. chain extenders also play a role in optimizing the processing conditions and curing times of the polymers, making them really important component in polymer synthesis.

For molecular extenders, the molecule is being chosen for its level of complexity, it provided a nice symmetrical structure contains most of the significant functional motifs that are critical in molecular backbones expansion, polymerization and molecular backbone's flexibility.

---

Prompt

---

I want you to think like a chemist performing a detailed analysis of the chemical composition of a chain extender through its constituent motifs. I will highlight for you 16 of the substructures of a molecule. They are numbered one-by-one from Motif 0 to Motif 15, inclusive. I want you to explain, concisely, what each numbered motif is. Make sure to start from Motif 0 and go in order of the numbering. MAKE SURE you describe EVERY MOTIF!

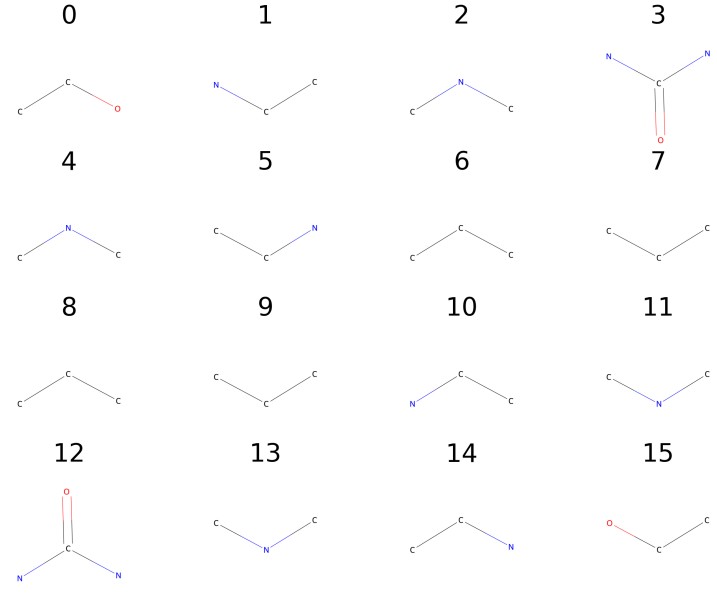

Response

Sure, I'll provide a concise description of each motif:

1. **Motif 0**: This structure is an ester functional group, specifically an alkyl chain with an ester linkage (C-O-C=O). 2. **Motif 1**: This is an imine (Schiff base) structure, featuring a double bond between carbon and nitrogen (C=N). 3. **Motif 2**: This is a secondary amine, consisting of a carbon-nitrogen single bond (C-N-C). 4. **Motif 3**: This is a nitrile group with an adjacent carbonyl, forming a structure known as a cyanoformamide (N-C=O). 5. **Motif 4**: This is a secondary amine with two different R groups (N-C-C). 6. **Motif 5**: This is again a secondary amine but with a different arrangement compared to Motif 2 (C-C-N). 7. **Motif 6**: This is a linear aliphatic chain fragment (C-C-C). 8. **Motif 7**: This is a linear N-C-C-N fragment featuring two secondary amines. 9. **Motif 8**: This is a simple three-carbon aliphatic chain (C-C-C). 10. **Motif 9**: This is another simple three-carbon aliphatic chain (C-C-C). 11. **Motif 10**: This is another linear aliphatic fragment, similar to Motif 8 and 9 (C-C-C). 12. **Motif 11**: This is a linear di-N-C-C fragment featuring two secondary amines. 13. **Motif 12**: This is a cyanoformamide (N-C=O) motif similar to Motif 3 but with two nitrogen atoms. 14. **Motif 13**: This is a linear N-C-C motif featuring a single secondary amine. 15. **Motif 14**: This is another linear N-C-C motif featuring a single secondary amine, similar to Motif 13. 16. **Motif 15**: This structure is an ester functional group, similar to Motif 0 but arranged differently within the chain (C-O-C=O).

Comments by Expert

Rate of prompt: easy prompt, only need to explain what each motif is, the number and motifs are clearly labeled. Emphasis on describing every motif prevent GPT in providing general answers to similar structures all at once. GPt's answer: fully correct, GPT describe each motif accurately
Prompt

I want you to think like a chemist performing a detailed analysis of the chemical composition of a chain extender through its constituent motifs. I will highlight for you some of the distinctive fragments of a molecule. They are numbered from 0 and individually highlighted in RED. Focus ONLY on the substructure highlighted in red within each cell. Here is the descriptions for each substructure provided by an expert:

Sure, I'll provide a concise description of each motif:

1. **Motif 0**: This structure is an ester functional group, specifically an alkyl chain with an ester linkage (C-O-C=O). 2. **Motif 1**: This is an imine (Schiff base) structure, featuring a double bond between carbon and nitrogen (C=N). 3. **Motif 2**: This is a secondary amine, consisting of a carbon-nitrogen single bond (C-N-C). 4. **Motif 3**: This is a nitrile group with an adjacent carbonyl, forming a structure known as a cyanoformamide (N-C=O). 5. **Motif 4**: This is a secondary amine with two different R groups (N-C-C). 6. **Motif 5**: This is again a secondary amine but with a different arrangement compared to Motif 2 (C-C-N). 7. **Motif 6**: This is a linear aliphatic chain fragment (C-C-C). 8. **Motif 7**: This is a linear N-C-C-N fragment featuring two secondary amines. 9. **Motif 8**: This is a simple three-carbon aliphatic chain (C-C-C). 10. **Motif 9**: This is another simple three-carbon aliphatic chain (C-C-C). 11. **Motif 10**: This is another linear aliphatic fragment, similar to Motif 8 and 9 (C-C-C). 12. **Motif 11**: This is a linear di-N-C-C fragment featuring two secondary amines. 13. **Motif 12**: This is a cyanoformamide (N-C=O) motif similar to Motif 3 but with two nitrogen atoms. 14. **Motif 13**: This is a linear N-C-C motif featuring a single secondary amine. 15. **Motif 14**: This is another linear N-C-C motif featuring a single secondary amine, similar to Motif 13. 16. **Motif 15**: This structure is an ester functional group, similar to Motif 0 but arranged differently within the chain (C-O-C=O).

I want you to tell me if any two of them should be combined together to form a more meaningful substructure. Remember that chain extenders often contain repeating units like ether, ester, amide, or urethane groups, and the goal is to extend the molecular backbone. Your task is to highlight the primary functional groups of the molecule. Output a single pair of numbers if you think those two fragments should be combined, and a brief explanation why. If no such pairs exist, don't output anything.

### Response

**Pair to Combine: Motif 2 and Motif 4**

**Explanation:**

- Both motifs feature secondary amines. - When combined, they could form a segment of a polyamine chain which is a common segment in certain polymers, potentially contributing to the chain extension and backbone of the resulting polymeric material.

### Comments by Expert

2484
2485
2486
2487
2488

Easy prompt, asking if any of the two motif pairs should be combined, provided detailed instructions on what functional groups chain extenders often include for GPT as references. GPT's answer: motif 2 and 4, GPT answered it correctly, as both contains secondary amines and satisfy the goal to extend molecular backbones. The combined structure also features high stability.

Prompt

2489
2490
2491
2492
2493

I want you to think like a chemist performing a detailed analysis of the chemical composition of a chain extender through its constituent motifs. I will highlight for you some of the distinctive substructures of a chain extender. They are numbered from 0. Here are the textual descriptions of each motif:

Sure, I'll provide a concise description of each motif:

2494
2495
2496
2497
2498
2499
2500
2501
2502
2503
2504
2505
2506
2507
2508
2509

1. **Motif 0**: This structure is an ester functional group, specifically an alkyl chain with an ester linkage (C-O-C=O). 2. **Motif 1**: This is an imine (Schiff base) structure, featuring a double bond between carbon and nitrogen (C=N). 3. **Motif 2**: This is a secondary amine, consisting of a carbon-nitrogen single bond (C-N-C). 4. **Motif 3**: This is a nitrile group with an adjacent carbonyl, forming a structure known as a cyanoformamide (N-C=O). 5. **Motif 4**: This is a secondary amine with two different R groups (N-C-C). 6. **Motif 5**: This is again a secondary amine but with a different arrangement compared to Motif 2 (C-C-N). 7. **Motif 6**: This is a linear aliphatic chain fragment (C-C-C). 8. **Motif 7**: This is a linear N-C-C-N fragment featuring two secondary amines. 9. **Motif 8**: This is a simple three-carbon aliphatic chain (C-C-C). 10. **Motif 9**: This is another simple three-carbon aliphatic chain (C-C-C). 11. **Motif 10**: This is another linear aliphatic fragment, similar to Motif 8 and 9 (C-C-C). 12. **Motif 11**: This is a linear di-N-C-C fragment featuring two secondary amines. 13. **Motif 12**: This is a cyanoformamide (N-C=O) motif similar to Motif 3 but with two nitrogen atoms. 14. **Motif 13**: This is a linear N-C-C motif featuring a single secondary amine. 15. **Motif 14**: This is another linear N-C-C motif featuring a single secondary amine, similar to Motif 13. 16. **Motif 15**: This structure is an ester functional group, similar to Motif 0 but arranged differently within the chain (C-O-C=O).

2510
2511
2512
2513
2514

I want you to pick only ONE of these as the root motif most essential to its chemical profile. It should be the single most important motif the rest of the chain extender was built around. Remember that chain extenders often contain repeating units like ether, ester, amide, or urethane groups, which help extend the molecular backbone, so your selected root motif MUST contain one of these repeating units. If there are multiple such motifs, or one doesn't clearly stand out, just pick one of them. Give your answer as a single number. Explain your reasoning carefully.

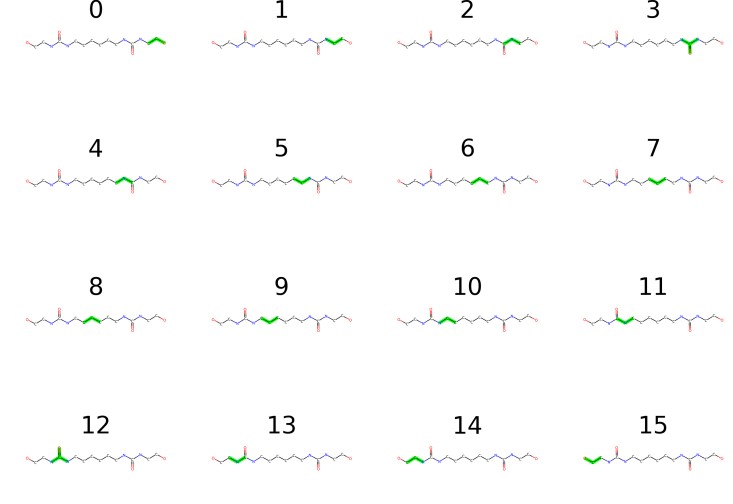

Response

2532
2533
2534
2535
2536
2537

After examining the given motifs, the root motif should possess a fundamental repeating unit pivotal for building the chain extender's molecular backbone. Esters, amides, and related functional groups are significant in forming backbones due to their capability to extend polymer chains. Among the given motifs, ester groups are notable candidates for this role.

Motif 0 and Motif 15 are both ester functional groups (C-O-C=O), which are commonly used as chain extenders due to their ability to form strong and flexible polymer chains. These motifs provide the necessary repetition for polymer chain growth and flexibility.

Since Motif 0 and Motif 15 are both relevant and appear similar, either could be considered. For simplicity and based on initial numbering, I will select **Motif 0**. This choice ensures the selection of a distinct ester functional group crucial to the overall chemical profile of the chain extender. Therefore, the single most essential motif for the chemical profile of the chain extender is:
**0**

---

Comments by Expert

---

Medium, this prompt let GPT pick the most important motif, which can sometimes result in discrepancies between the answers from language models and chemistry expert as some motifs may have similar levels of significance. GPT's answer : GPT answered motifs 0 and 15, and mentioned that the reason is because they contain the ester functional group that are necessary for polymer chain growth and the flexibility of the chain. This is correct, so GPT answered this question right. But from the perspective of a chemistry expert, motifs 3 and 12 are also important structure in chain extend molecules. Since this prompt only let GPT pick one, so the answer that GPT provided are still considered correct.

## A.6 CONCLUDING THOUGHTS

In summary, language models excel at performing direct and explicit tasks, such as describing the names and functionalities of chemical motifs. These types of questions typically have abundant training data, enabling language models to provide 100% accurate answers and descriptions of all chemistry motifs and their properties. Moreover, language models are highly effective at simple post-processing tasks, such as extracting key terms from answers, achieving 100% accuracy in such cases. On the other hand, when it comes to tasks requiring empirical knowledge of organic chemistry synthesis, such as identifying which motif is most important, the models show some limitations. However, in case studies, they still reach over 90% accuracy, reflecting the inherent ambiguity in experiential knowledge. Surprisingly, the language models perform exceptionally well in evaluating the importance of interactions between motifs, often providing nearly perfect responses. This indicates that the models excel at learning molecular structure-property relationships by studying molecular graphs, further proving that this methodology has broad potential and numerous applications.

Figure 7: Input molecule from Isocyanates

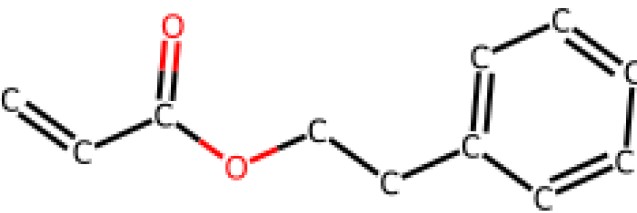

Figure 8: Input molecule from Acrylates

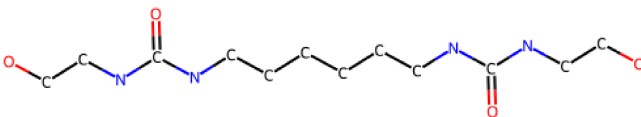

Figure 9: Input molecule from Chain Extenders