# OpenReview forum: "Multi-Modal Foundation Models Induce Interpretable Molecular Graph Languages"
_ICLR.cc/2025/Conference — ICLR 2025 Conference Withdrawn Submission_

### Official Review · Reviewer_B8ZR · 2024-10-29

**Soundness:** 2
**Presentation:** 1
**Contribution:** 2
**Rating:** 3
**Confidence:** 3

**Summary:**

This work employs a multimodal language model (MLLM) as a decision-maker in the molecular graph language learning process. In this procedure, each molecule is rendered as an image for MLLM input, and the model outputs decisions and descriptions based on specific prompts. The resulting learned molecular grammar is then applied to generate new molecules within certain classes, demonstrating strong performance.

**Strengths:**

1. Utilizing MLLM as a decision-maker in the molecular tree composition process is a strong approach, and using rendered molecular images as input is a clever choice.

2. The experimental results appear to perform as well as domain expert annotations.

**Weaknesses:**

Overall, I think the use of MLLM as a decision-maker in the graph grammar learning, or “tree decomposition graph construction” process, is promising. However, the paper’s presentation and writing lack clarity, making it difficult to follow and understand. Additionally, many critical experimental details are missing, which limits the reproducibility and applicability of the method.

1. Lack of Definitions: In the abstract, terms like MMFMs and DSLs are introduced without explanation. I suspect that these abbreviations are under-defined. In the methods section, it would help if the authors included explanations or examples of these terms.

2. Lack of Structure: This method appears aimed at addressing a molecular domain-specific language learning task. However, the introduction section offers no information about molecular language, which surprisingly only appears in the Related Work section. This organization feels unusual and somewhat illogical.

3. Lack of Model and Experimental Details: Both the methods and experiments sections lack fundamental details. For example, which MMFM does this approach employ? What prompts are specifically used? What is the dataset description and training cost? How are the baselines evaluated? I am particularly curious about the training and inference procedures, as the method seems to rely on MLLMs to decide the tree decomposition construction of clique graphs, yet it’s unclear how this process is applied to generate new molecules. Was fine-tuning involved, or was it entirely prompt-based?

**Questions:**

In terms of weaknesses, I find it challenging to fully understand or verify the details of the proposed method. The current lack of clarity and the absence of key methodological details make it difficult to assess the approach’s validity and potential for replication. I strongly believe that the paper requires substantial revision to address these gaps. Adding detailed explanations and structural improvements would better support the work’s contributions, as it currently does not seem ready for publication.

---

### Official Review · Reviewer_nzt1 · 2024-10-30

**Soundness:** 2
**Presentation:** 1
**Contribution:** 1
**Rating:** 1
**Confidence:** 3

**Summary:**

This paper explores the potential of multi-modal foundation models (MMFMs) to craft Domain Specific Languages (DSLs) for chemistry. The key argument is that DSLs are very useful, and it's a good idea to build DSLs on specific domains as they facilitate rules for better explaining decisions from models, in this case the decoding process they follow allows them to generate molecules while also providing explainations. This is useful because domain-experts typically trust more something they can rationalize.
The authors finally show the performance of their method on some molecular generation benchmarks.

**Strengths:**

The work is certainly original in their use of MMFMs for describing molecular substructures and then using them again for proposing how to step-by-step build molecules from those motifs. The idea here is that the MMFM can guide and rationalize the generation of molecules in a given subfield.

The authors make a good point that LLMs lack abilities to understand chemical objects such as reactions and molecules, especially when these are given in SMILES format which is the most common thing to use, as graphs cannot be directly fed into LLMs. However images depicting the molecules and other things are a good idea to elicit correct chemical analyses from MMFMs, and it seems to work well to describe motifs, molecules, and perform other tasks such as suggesting combinations of motifs.

**Weaknesses:**

### Writing

- It is not very clear what the goal of the paper is. Is it molecular generation, or DSL generation? In either case, very little insight is given into how the generated molecules look like, or how the generated DSL looks like. This is important provided that the paper is so strongly focused on applications.

### Potentially false or misleading claims, lack of evidence/citations.
- In general the whole introduction section misses a lot of citations. Most of the claims made there are not based on evidence, excepting 3 citations on popular LLM papers, and 1 (Makatura, 2023) that works on LLMs for aid in design.
- Section 2.3, where the role of FMs for molecular generation is discussed. The authors make several claims that are either false or misleading:
  - "SMILES or SELFIES are mainly for representation purposes and can lead to issues in the context of generation". The SELFIES system was specifically designed for molecular generation, one of the advantages being that every SELFIES string represent a valid molecule, tackling any concerns regarding validity [1].
  - The authors state that the alternative to FMs for molecular generation are "GNNs or million-parameter language models for text" which "require extensive training resources". No evidence or citation is provided for this, and furthermore the current work presents no analysis of the computational resources used by the presented method.
  - The state of the art for molecular generation are indeed language models trained on SMILES [2-4]. Regarding the computational efficiency of these methods, there's a lot of active research focusing on improving the sample efficiency of these methods [5], however none of these works has been considered when making the claims above, nor does the work compare against them in any way.

### Results
- It is not clear in the results section where each result is coming from, as no citation is linked to each of the methods listed.
- The notation is very unclear in Table 1 and 2. In particular, the notation Isocyanates (11), does it mean that the dataset of Isocyanates contains 11 samples? This is not clearly stated. Are the results aggregated from the dataset containing Isocyanates, Acrylates and chain extenders? why is this dataset designed like that?
- It's very unclear what each column represents in these tables. The caption should at least specify this.
- The analysis is not clear. Example "...methods do better on 3), but struggle across dimensions 2) and 3).", what is meant by "2)" and "3)"? is it refering to Novelty and Diversity? this is not clear and never stated
- "However, FMG still leaves some to be desired across 3)." this sentence is not clear.
- "FMG appears to do exceptionally well for PTC (halides) but poor for HOPV (thiophenes), which is surprising considering. As we..." this sentence is incomplete? "which is surprising considering...?"


### References
[1] Krenn, M., Häse, F., Nigam, A., Friederich, P., & Aspuru-Guzik, A. (2019). SELFIES: a robust representation of semantically constrained graphs with an example application in chemistry. arXiv preprint arXiv:1905.13741, 1(3).
[2] Blaschke, T., Arús-Pous, J., Chen, H., Margreitter, C., Tyrchan, C., Engkvist, O., ... & Patronov, A. (2020). REINVENT 2.0: an AI tool for de novo drug design. Journal of chemical information and modeling, 60(12), 5918-5922.
[3] Öztürk, Hakime et al. “Exploring Chemical Space using Natural Language Processing Methodologies for Drug Discovery.” Drug discovery today (2020)
[4] Özçelik, R., de Ruiter, S., Criscuolo, E. et al. Chemical language modeling with structured state space sequence models. Nat Commun 15, 6176 (2024). https://doi.org/10.1038/s41467-024-50469-9
[5] Guo, J., & Schwaller, P. (2023). Augmented memory: Capitalizing on experience replay to accelerate de novo molecular design. arXiv preprint arXiv:2305.16160.

**Questions:**

1. Can you provide more details or examples of the generated molecules and DSL? This would help readers better understand the practical outcomes of your method.

2. The introduction lacks citations for many of the claims made. Could you provide evidence or references to support these statements, particularly regarding the challenges and current state of molecular generation?

3. Regarding your claims about SMILES and SELFIES in Section 2.3, could you address the fact that SELFIES was designed specifically for molecular generation and ensures valid molecules? How does this impact your argument?

4. You mention that alternatives to FMs for molecular generation require extensive training resources. Can you provide evidence or comparisons to support this claim, particularly in relation to your method's computational requirements?

5. Could you clarify the notation used in Tables 1 and 2, particularly the meaning of numbers in parentheses (e.g., Isocyanates (11))? What do these represent?

6. In the results section, can you provide citations for each of the methods listed and clarify what each column in the tables represents?

7. Your analysis refers to dimensions "2)" and "3)" without clear explanation. Could you elaborate on what these refer to and how they relate to the metrics presented?

8. There seems to be an incomplete sentence in your analysis: "FMG appears to do exceptionally well for PTC (halides) but poor for HOPV (thiophenes), which is surprising considering." Could you complete this thought?

9. Have you considered comparing your method against state-of-the-art language models trained on SMILES for molecular generation? How does your approach compare in terms of efficiency and effectiveness?

10. Can you discuss how your work relates to recent research on improving sample efficiency in molecular generation?

11. Could you elaborate on the details of the method? e.g. what temperature was used for gpt-4o, what image parameters were used (e.g. image resolution, size, etc). Does any of these variables have any influence on the results?

12. For Table 1 and 2, it's not clear how many molecules were generated. Could you please specify this?

---

### Official Review · Reviewer_vnXU · 2024-11-03

**Soundness:** 2
**Presentation:** 2
**Contribution:** 2
**Rating:** 5
**Confidence:** 4

**Summary:**

Through this paper, the authors propose Foundation Molecular Grammar (FMG), a method that constructs domain-specific languages (DSLs) in a data-efficient manner using multi-modal foundation models (MMFMs). Specifically, FMG eases the MMFM’s task by casting the DSL construction into the problem of constructing a tree decomposition for the molecular graph.

**Strengths:**

- Overall, the paper was easy to follow. The writing and the concept figure were clear.
- An ablation study was conducted for the MMFM module.

**Weaknesses:**

I will combine the *Weaknesses* section and the *Questions* section. My concerns are as follows:
- Some abbreviations are used without explanation of the full term. For example, the full term for DSL, FM, and MMFM should be provided in Introduction. The full term for the proposed method, FMG, is also only in Abstract and not in the main text.
- The main weakness of this paper is that the experiments are not extensive and robust. Why only grammar-based and VAE methods were selected as a baseline out of the vast molecular generative methods? Moreover, only small and medium datasets were used in the experiments. It would be great to provide results using more popular and larger datasets such as ZINC250k or MOSES for a broader comparison with previous methods.
- Interpretability is a major advantage of the proposed method, but this advantage is not properly explained and emphasized in the experiment section. I strongly recommend devoting a few paragraphs to interpretability of FMG with a case study.
- The authors did not provide the codebase to reproduce the results.

**Questions:**

Please see the *Weaknesses* section for my main concerns.

For now, I’m leaning toward borderline reject, but I’ll be glad to raise the score when all the questions are fully addressed.

---

### Official Review · Reviewer_DkB4 · 2024-11-06

**Soundness:** 3
**Presentation:** 3
**Contribution:** 2
**Rating:** 6
**Confidence:** 3

**Summary:**

This paper proposes a method to induce a DSL for building molecules from a given subdomain by casting the DSL construction as a sequence of steps and using a large multimodal pretrained model to make those intermediate choices. The authors then show promising results on a few relevant molecule classes.

**Strengths:**

(S1): The paper is highly novel, exploring a quite unusual research direction. The writing is relatively clear and easy to follow.

(S2): Apart from providing the main experiments, the authors also ablate their method quite thoroughly, replacing all the FM-based components with reasonable heuristics.

**Weaknesses:**

(W1): I am not sure if the main experiments in this work are representative of real-world use. Is being able to simply generate/sample molecules from a given subdomain useful in itself, or would it only be useful if paired with molecular optimization?

(W2): It's not clear to me how the VAE baselines are set up. Are these models pretrained and then fine-tuned on the (small) dataset in question, or trained on the latter directly? Would it make sense to instead use a frozen pretrained VAE and steer it to sample around a given subdomain by inferring the right region of latent space to sample from? Alternatively, for motif-based models such as HierVAE, one could also constrain the set of motifs to those that appear in the given dataset describing the domain.

=== Other comments ===

In the line of VAE-based models there's also MoLeR (from "Learning to Extend Molecular Scaffolds with Structural Motifs"), which is a more modern extension of JT-VAE/HierVAE, often shown to perform better than the latter.



=== Nitpicks ===

Below I list nitpicks (e.g. typos, grammar errors), which did not have a significant impact on my review score, but it would be good to fix those to improve the paper further.

- Top of page 3: "notations like SMILES or SELFIES are mainly for representation purposes and can lead to issues (…). This may hinder LLMs’ understanding as they lack sufficient pre-training on these notations compared to SMILES" is confusing

- Lines 189-191: it's not clear how "u, v share an atom" should be interpreted given that context suggests u and v are atoms/nodes?

- Line 403: "We first observe in Tables 1 and that" - something is missing

- Line 407: the authors refer to "dimensions" without explanation of what this means (I assume each dimension is one of the datasets?)

- Line 426: "surprising considering." - something is missing

**Questions:**

See the "Weaknesses" section above for specific questions.

---

### Note · Authors · 2024-12-01

I have read and agree with the venue's withdrawal policy on behalf of myself and my co-authors.